# High temperatures alter cross-over distribution and induce male meiotic restitution in *Arabidopsis thaliana*

Nico De Storme[1,2] & Danny Geelen [1✉]

Plant fertility is highly sensitive to elevated temperature. Here, we report that hot spells induce the formation of dyads and triads by disrupting the biogenesis or stability of the radial microtubule arrays (RMAs) at telophase II. Heat-induced meiotic restitution in *Arabidopsis* is predominantly SDR-type (Second Division Restitution) indicating specific interference with RMAs formed between separated sister chromatids. In addition, elevated temperatures caused distinct deviations in cross-over formation in male meiosis. Synapsis at pachytene was impaired and the obligate cross-over per chromosome was discarded, resulting in partial univalency in meiosis I (MI). At diakinesis, interconnections between non-homologous chromosomes tied separate bivalents together, suggesting heat induces ectopic events of non-homologous recombination. Summarized, heat interferes with male meiotic cross-over designation and cell wall formation, providing a mechanistic basis for plant karyotype change and genome evolution under high temperature conditions.

[1] Department of Plants and Crops, Faculty of Bioscience Engineering, Ghent University (UGent), Coupure Links 653, 9000 Ghent, Belgium. [2] Present address: Laboratory for Plant Genetics and Crop Improvement (PGCI), Division of Crop Biotechnics, Department of Biosystems, KU Leuven, Willem de Croylaan 42, 3001 Heverlee, Leuven, Belgium. ✉email: Danny.Geelen@ugent.be

One of the prevailing effects of high temperatures on crop production is the reduction of reproductive success[1]. The male reproductive system has been repeatedly found to be most susceptible to temperature stress[2]. Brief as well as prolonged shifts in the ambient temperature promote 2n gamete formation in a broad range of species, including *Lotus tenuis*[3], wheat[4], rose[5], *Brassica*[6], *Arabidopsis*[7], *Dianthus caryophyllus*[8], and *Populus*[9,10]. However, the underlying cellular defect and genetic outcome of meiotic nonreduction greatly varies depending on species and temperature regime. In *Arabidopsis thaliana*, cold stress induces male meiotic restitution by specifically interfering with the RMAs that underpin biogenesis of the second cell wall at the end of MII, and hence yields dyads and triads[7]. Similarly, in wheat and poplar, MII cell wall formation is affected by heat, indicating that the process of male meiotic cytokinesis is highly temperature-sensitive[10,11]. In rose, heat-induced meiotic restitution is caused by alterations in MII spindle organization. Whereas the two MII syncytial spindles normally exhibit a perpendicular orientation, heat-stressed PMCs in rose often display a parallel, fused, or tripolar orientation of MII spindles, yielding restituted dyads and triads instead of regular tetrads at the end of MII[5]. Similar alterations in MII spindle orientation also underpin heat-induced male meiotic restitution and the associated production of 2n gametes in other plant species, such as poplar[10] and *Brassica ssp*[12]. Overall, these studies bring forward the cytoskeleton, and particularly microtubule (MT)-based arrays, in male MII as temperature-sensitive structures that undergo depolymerization and incomplete restoration upon heat or cold to cause alterations in MII chromosome dynamics and meiotic restitution[7].

From a genetic point of view, cellular mechanisms of diploid (2n) gamete formation can be subdivided into two types: first division restitution (FDR) and second division restitution (SDR)[13,14]. FDR-type mechanisms produce 2n gametes that are equivalent to those resulting from loss of the first meiotic cell division, either with or without homologous recombination. As a result, FDR-type 2n gametes contain non-parental sister chromatids and thus maintain parental heterozygosity at regions close to the centromere. In contrast, SDR-type mechanisms produce 2n gametes that are equivalent to those resulting from an omission of the second meiotic cell division (MII). The resulting SDR-type 2n gametes maintain both sister chromatids and thus are homozygous at regions close to the centromere.

Importantly, heat also affects the spatial–temporal dynamics of cross-over designation during meiosis I, leading to strong shifts in the genome-wide cross-over landscape. In barley, for example, heat causes an overall increase in cross-over rate together with a redistribution of crossovers from distal toward more proximally located chromosome regions[15,16]. In *Arabidopsis*, both heat and cold increase cross-over formation via the specific type I interference-sensitive cross-over pathway[17]. Little is yet known about the plasticity and dynamic behavior of cross-over designation under varying temperatures as well as on the underlying regulatory mechanisms. Initial studies indicate for a putative mechanistic control via the chromosome axis or the synaptonemal complex (SC)[16,18]; however, further studies are needed to clarify the molecular mechanisms underpinning temperature-induced shifts in cross-over distribution.

Heat stress has pleiotropic effects on male sporogenesis, including aberrations in CO distribution, MII cell division, and meiotic chromosome dynamics. Here, we report on a genetic and cytological analysis of the impact of heat stress on *Arabidopsis thaliana* male meiosis and show an array of defects in crossing over and cytokinesis. Heat stress interferes with RMA biogenesis to cause SDR-type meiotic restitution and 2n pollen production. Male meiosis additionally exhibits structural alterations in synaptonemal buildup together with the associated alterations in CO obligation, distribution, and fidelity. Overall, with heat-induced meiotic defects leading to the stochastic formation of di- and aneuploid spores, we here postulate that meiotic plasticity under temperature stress may constitute a mechanistic basis driving plant genome evolution and karyotype change under environmentally challenging conditions.

## Results

**Heat stress induces restitution of *Arabidopsis* male meiosis.** The impact of heat on male meiosis was investigated by exposing flowering *Arabidopsis* plants for 0, 12, 24, 36, and 48 h to mild (26–28 °C) and more extreme temperature (30–32 °C). Flower buds carrying tetrad-stage male meiocytes were microscopically analyzed immediately after heat treatment. Under standard conditions (18–20 °C), *Arabidopsis* male sporogenesis always yields regularly shaped tetrads, i.e., typically composed of four uniformly sized spores that are arranged in a tetrahedral configuration (Fig. 1a). The uniform tetrahedral structure reflects the tight three-dimensional control of chromosome segregation and nuclear organization in male MI and MII, and hence can be used as readout for putative alterations in one of these meiotic processes. Under mild heat (26–28 °C), *Arabidopsis* male meiosis primarily yields balanced tetrads (~95%) and a minor fraction of triads, dyads, monads, and polyads, indicating for defects in male meiotic cell division (Supplementary Fig. 1). Among the dyads, two different types were distinguished: balanced dyads with two equally sized (2× + 2×) spores (Fig. 1c) and unbalanced dyads, in which one spore is significantly larger than the other one (3× + 1×). Whereas unbalanced dyads can result from a range of different meiotic alterations, balanced dyads typically result from a defect in cytokinesis or cell wall formation. Meiotic polyads (Fig. 1d), on the other hand, contain more than four spores, and typically result from alterations in MI or MII chromosome dynamics.

Incubating flowering plants at higher temperatures (30–32 °C) for 12 h or longer increases the frequency of the combined set of altered male meiotic products above 20% (Fig. 2). Interestingly, the proportion of polyads (15–30%) appears consistently much higher under this temperature regime compared with the mild heat stress treatment (0–10%) (Fig. 2). Moreover, under these temperature conditions (30–32 °C), male meiosis additionally yields unbalanced tetrads that typically carry two small and two larger microspores (Fig. 1p). These unbalanced tetrads are rare under mild heat stress (26–28 °C), indicating that depending on the temperature range, different meiotic processes are affected.

For both mild and moderate heat, there was often variation in the frequency of different types of altered meiotic products between plants, putatively reflecting differences in the exact meiotic stage at which the analyzed flower buds were exposed. However, no differences were observed in the total number of altered meiotic products (Fig. 2), indicating that *Arabidopsis* male meiosis exhibits a consistent sensitivity to the ambient temperature environment.

To investigate whether high-temperature stress affects premeiotic processes and whether heat-triggered aberrant microspores develop into viable pollen, we next assessed different stages of male gametogenesis in a *qrt1–2*$^{-/-}$ background at different time intervals (e.g., 0, 12, 24, and 36 h) following heat exposure (24 h at 30–32 °C). Whereas tetrad-stage meiocytes immediately after heat show clear structural alterations, no such defects were observed at the meiotic tetrad stage 12 h following heat and later (Supplementary Fig. 2). This indicates that heat does not interfere with premeiotic PMC development and does not have a persistent imprinting effect on male meiotic cell division. Moreover,

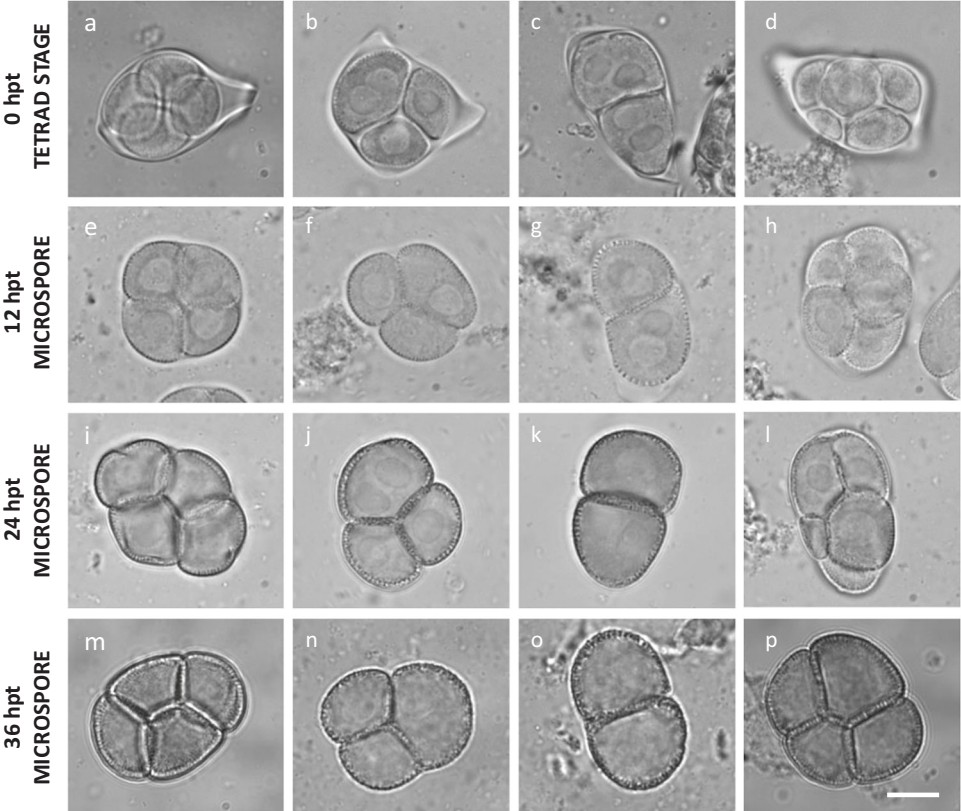

**Fig. 1 Heat induces defects in *Arabidopsis* male meiotic cell division.** Representative images of lactopropionic orcein-stained tetrad-stage male meiocytes (**a–d**) and early-, mid-, and late-stage microspores (**e–p**) of *Arabidopsis qrt1–2*$^{-/-}$ plants grown under normal temperature conditions (18–20 °C) (**a**, **e**, **i**, **m**) and at different time periods following heat treatment (24 h at 30–32 °C) (**b–d**, **f–h**, **j–l**, **n–p**). Hpt means hours post treatment. Scale bar, 10 μm.

microscopic analysis of *qrt1–2*$^{-/-}$ microspore stages at different periods following a 24-h 30–32 °C heat exposure revealed the presence of progressively developed dyads, triads, and polyads at more advanced stages of microgametogenesis (Fig. 1), indicating that altered meiotic products generated by heat proceed into gametophytic development. Further analysis revealed that, at 7–8 days post heat treatment, altered *qrt1–2*$^{-/-}$ spore configurations, including monads, dyads, and triads, occur in the mature pollen grain stage (Supplementary Fig. 2). Altogether, these findings demonstrate that the altered meiotic products formed under heat do not arrest in development and enter into the microgametogenesis program to develop into mature pollen grains.

**Heat-induced male meiotic restitution is caused by defects in MII cell wall formation.** The combined formation of dyads, triads, and monads generally indicates for alterations in MII cytokinesis. To validate whether this occurs under heat, meiotic cell plate formation at the end of MII was monitored using anilin blue staining. Under normal temperature conditions (18–20 °C), tetrad-stage PMCs ($n = 155$) consistently show a tetrahedral configuration of meiotic nuclei with a distinct X-shaped callosic cell wall that physically isolates all four haploid nuclei from each other (Fig. 3a). Upon heat exposure (24 h at 30–32 °C), most meiocytes exhibit an X-shaped callosic cell wall pattern similar to wild type; however, in some PMCs (22%: $n = 236$), one or more internuclear cell walls are absent or reduced to stubs, to consequently yield meiotically restituted figures (Fig. 3b–h). Interestingly, early-stage microspores that were exposed to heat during meiosis remain ectopically attached to each other and reveal strong accumulation of callose at the contact site (Fig. 3i–p).

Hence, deposition and subsequent degradation of internuclear callose in tetrad-stage *Arabidopsis* male meiocytes are impaired by high-temperature stress.

Callose deposition at the boundary of nuclear cytoplasmic domains (NCDs) at the end of MII relies on prior establishment of internuclear phragmoplast-like structures, called radial microtubule arrays (RMAs). In order to assess the biogenesis and stability of RMAs and other meiotic microtubule (MT) structures under heat, we performed β-tubulin immunostaining of male meiocytes. Similar as under control conditions, heat-stressed PMCs generate a distinct perinuclear MT array at prophase I, show polarly oriented spindles at metaphases I and II, and produce a distinct phragmoplast MT array between the two polar chromosome sets at interkinesis (Supplementary Fig. 3). At the end of MII, however, heat-stressed PMCs exhibit clear alterations in RMA biogenesis (Fig. 4). Contrary to control male meiocytes, which exhibit the consistent formation of MT arrays between each of the four haploid nuclei at telophase II (Fig. 4a–c), a subset of heat-stressed PMCs (~25%, $n = 83$) displays incomplete, aberrant, or absent RMAs, i.e., as reflected by the lack of MTs between two or more nuclei (Fig. 4d–l). Moreover, although most heat-stressed PMCs harbor four distinct nuclei of uniform size, some PMCs (±5%) were found to contain five or more nuclei of a variable size, irrespective of alterations in RMA organization (Fig. 4j–l). Meiotic products with extra nuclei correspond to MII polyad formation and provide additional evidence for the occurrence of defects in MI or MII chromosome segregation under heat. Both defects in RMA biogenesis and chromosome segregation occur independently of each other and thus reflect the presence of multiple heat-sensitive elements that separately operate in different meiotic stages or processes.

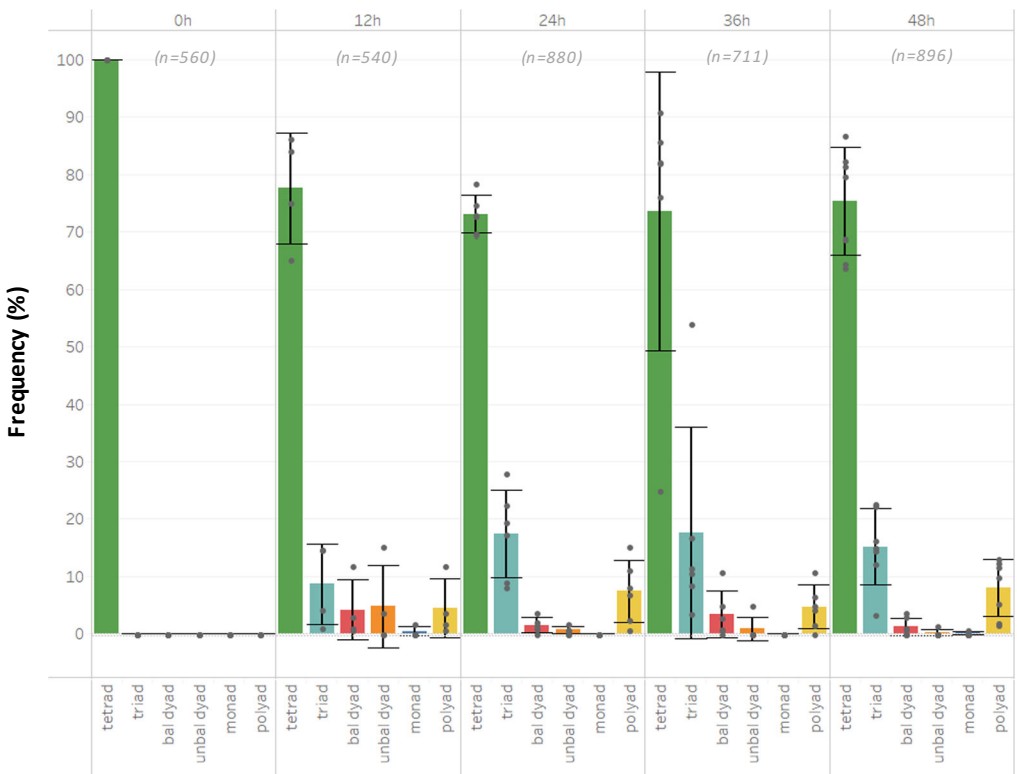

**Fig. 2 *Arabidopsis thaliana* male meiosis exposed to heat (30–32 °C) yields dyads, triads, and polyads.** Quantitative analysis of the different types of male meiotic figures produced by *Arabidopsis thaliana* 2× Col-0 male sporogenesis exposed to varying periods of moderate heat stress (30–32 °C). Values represent the mean of >500 male meiotic products isolated from at least three different plants. The total number of male meiotic products analyzed for each specific treatment is indicated by value *n*. Error bars represent standard deviation values.

**Syncytial nuclei in bi- and polynuclear spores fuse before pollen mitosis I.** In order to assess postmeiotic development and ploidy stability of bi- and polynuclear spores resulting from heat-stressed PMCs, centromere dynamics during microspore development was assessed using the pWOX2-CENH3-GFP reporter. Under standard temperature conditions (18–20 °C), early-stage microspores consistently contain one single fluorescent nucleus (Fig. 5a, *n* > 200), whereas upon heat stress ±25% of the spores harbor two to four syncytial nuclei (Fig. 5b, c, *n* > 100). Three days post heat treatment, spores with multiple nuclei reveal distinct GFP foci in each individual nucleus, representing centromeric regions. The loading of CENH3-GFP onto the centromeres indicates that bi- and polynuclear spores enter into the mitotic cell cycle and thus proceed into microgametophytic development. Individual nuclei in bi- and polynuclear spores at 3 days post treatment generally display five centromeric GFP foci (Fig. 5e, f, *n* = 25), reflecting a regular haploid chromosome number. However, a subset of enlarged spores contains only one nucleus with ten GFP foci (Fig. 5h, i, *n* = 4), indicative for a diploid state as a result of the fusion of two syncytial haploid nuclei. At 4 days post heat treatment, enlarged unicellular-stage microspores consistently harbor one enlarged nucleus with 10 or 15 centromeric dots (Fig. 5k, l, *n* = 17) suggesting for the fusion of two or three haploid nuclei, respectively. In the next developmental stage, i.e., following pollen mitosis I (PMI), enlarged spores consistently carry two gametophytic nuclei, i.e., one less-condensed vegetative and one highly condensed generative nucleus with the latter one showing 10 or 15 centromeric GFP dots (Fig. 5n, o, *n* = 9). From this, we conclude that di- and polynuclear spores resulting from heat-stressed meiosis show nuclear fusion before PMI to yield di- and polyploid gametes, respectively.

To monitor whether heat-induced bi- and polynuclear spores eventually develop into mature, tricellular pollen grains, nuclear chromatin was assessed in sperm cells using the pMGH3-H2B-GFP marker. Enlarged pollen observed at 7 and 8 dpt consistently carry two equally sized fluorescent sperm cells that appear larger than those in pollen resulting from meiocytes that developed under normal temperatures (Fig. 5p–w). This indicates for the formation of normally configured di- and polyploid pollen grains, and supports the finding that heat-induced male meiotic restitution leads to functional higher-ploidy gametes.

**Heat-induced male meiotic restitution primarily generates SDR-type 2n pollen grains.** Diploid gametes formed by meiotic restitution either maintain parental heterozygosity at the centromere (FDR, first division restitution) or instead are homozygous at the centromeric region (SDR, second division restitution). In order to assess which type of 2n gametes is formed upon heat exposure, two *Arabidopsis* fluorescent-tagged lines (FTL, CEN3 and CEN5) that each harbor two centromere-linked pollen-specific reporters in the *qrt1-1⁻/⁻* background (Fig. 6c) were subjected to heat stress (36 h, 30–32 °C) after which fluorescence segregation patterns were monitored in dyad and triad pollen figures at 7–9 dpt (Fig. 6a). Quantification of the described fluorescence patterns in heat-induced dyads and triads using the different centromere-linked FTL markers revealed that 76% of heat-induced restituted PMCs are SDR-type, whereas the remaining ones are considered FDR-type (Fig. 6b; Supplementary Table 1). FTL-based FDR/SDR analysis, however, assumes a complete absence of meiotic recombination between the genomic position of the fluorescent reporter and the centromere, potentially implying that the frequency of SDR-type restitution is underrepresented.

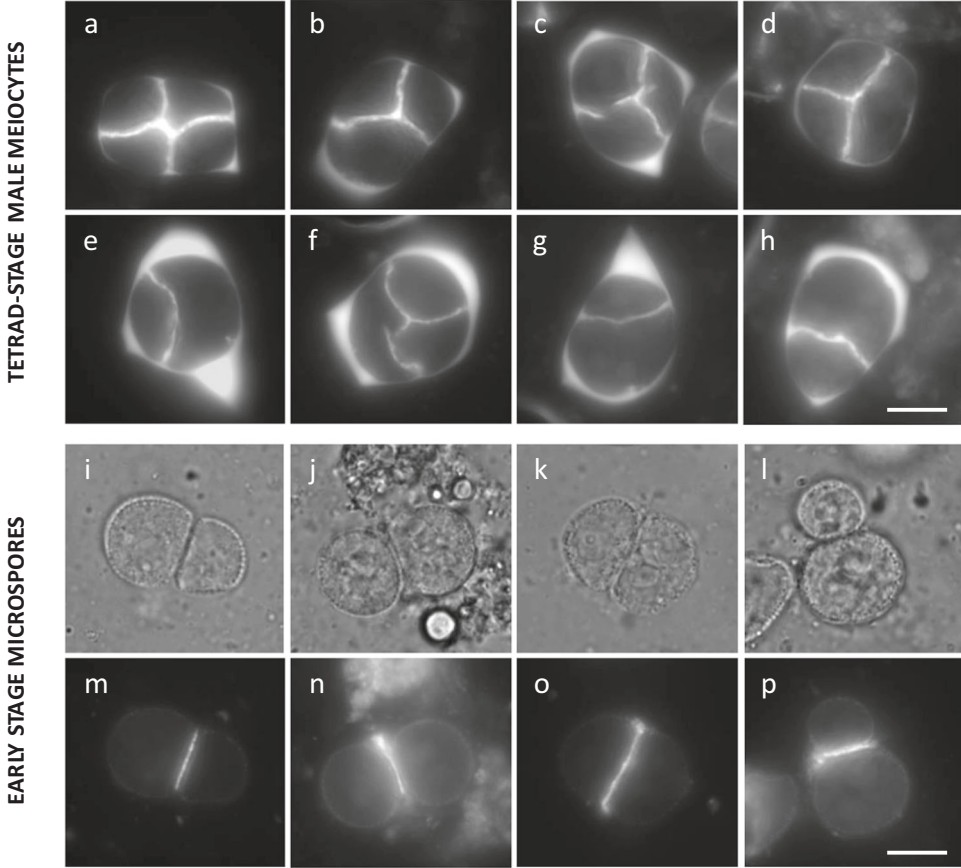

**Fig. 3 *Arabidopsis* male meiosis exposed to heat shows defects in MII cell wall formation. a–h** Anilin blue staining of tetrad-stage male meiocytes of diploid *Arabidopsis* Col-0 plants under normal temperature conditions (18 °C, **a**) and upon exposure to heat (24 h at 30–32 °C, **b–h**). **i–p** Anilin blue staining of early-stage microspores originating from heat-stressed *Arabidopsis* PMCs as viewed via bright-field (**i–l**) and fluorescent microscopy (**m–p**). Scale bar, 10 μm.

To test for the occurrence of residual FDR-type restitution under heat, tetrad analysis in *Arabidopsis* Col-0 haploid plants was performed, both upon heat and under normal conditions. Haploid plants only have one copy of each chromosome ($2n = 1x = 5$) and are devoid of meiotic recombination and bivalent formation. As a result, haploid meiosis I generates five univalents that randomly segregate in MI to yield two unbalanced sets of chromosomes that subsequently segregate in MII to produce a mix of unbalanced tetrads and polyads. FDR-type restitution of haploid meiosis nullifies unbalanced segregation in MI and yields balanced dyads, whereas SDR-type restitution maintains unbalanced separation of chromosomes in MI and therefore produces unbalanced dyads. Under normal conditions, male meiosis in haploid plants predominantly yields unbalanced tetrads (82.6% ± 7.0%) together with a minor subset of polyads (7.8% ± 3.9%), triads (2.6% ± 2.6%), and balanced dyads (7.1% ± 2.4%) (Fig. 6d, e). Upon exposure to heat (36 h at 30–32 °C), haploid PMCs produce a significantly lower number of unbalanced tetrads (12.2% ± 1.9%) and polyads (1.1% ± 0.5%), but instead produce more triads (33.4% ± 8.4%), unbalanced dyads (25.8% ± 4.8%), and monads (9.1% ± 3.9%) (Fig. 6d, e). The latter is in line with the predominant occurrence of SDR-type restitution under heat. Importantly, haploid PMCs exposed to heat also show a significant increase in the relative number of balanced dyads (18.4% ± 2.0%) (Fig. 6e), confirming that a subset of heat-induced meiotic restitution events (14.7%) occurs via an FDR-type mechanism.

Quantitative analysis of meiotic restitution was also performed in the *Arabidopsis* atps1–1$^{-/-}$ mutant. AtPS1 regulates

perpendicular orientation of the two MII spindles in male meiosis, with atps1$^{-/-}$ mutants typically exhibiting ectopic tripolar, parallel, and fused spindles in meiosis II. Under normal temperature conditions, atps1$^{-/-}$ male meiosis yields a mixture of tetrads (27.5% ± 8.0%), triads (28.7% ± 8.0%), and balanced dyads (42.7% ± 5.1%) (Supplementary Fig. 4), reflecting alterations in MII spindle orientation. Upon heat shock (36 h at 30–32 °C), atps1$^{-/-}$ male meiosis produces relatively less tetrads and triads, but instead yields a significantly increased number of balanced dyads (Supplementary Fig. 4), indicating for additional FDR-type restitution through loss of the first meiotic cell wall. Although heat-stressed atps1$^{-/-}$ does not yield unbalanced dyads, the occurrence of monads indicates that a minor part of heat-induced restitution events in the atps1–1$^{-/-}$ background is SDR-type. Hence, contrary to the predominant SDR-type restitution in the case of normal MII chromosome dynamics, heat predominantly causes FDR-type restitution in meiocytes with a parallel spindle configuration in MII.

**Heat stress causes partial asynapsis and interferes with chiasmata formation in male meiosis I.** Occurrence of polyads in heat-stressed PMCs indicates for defects in MI or MII chromosome segregation. To assess this in more detail, chromosome dynamics in male meiosis was cytologically analyzed in plants that were exposed for a short (6 h) or prolonged time (24 h) to heat (30–32 °C), i.e., to discriminate between direct and secondary effects. Under control temperatures, prophase I displays a progressive condensation and pairing of chromosomes to yield

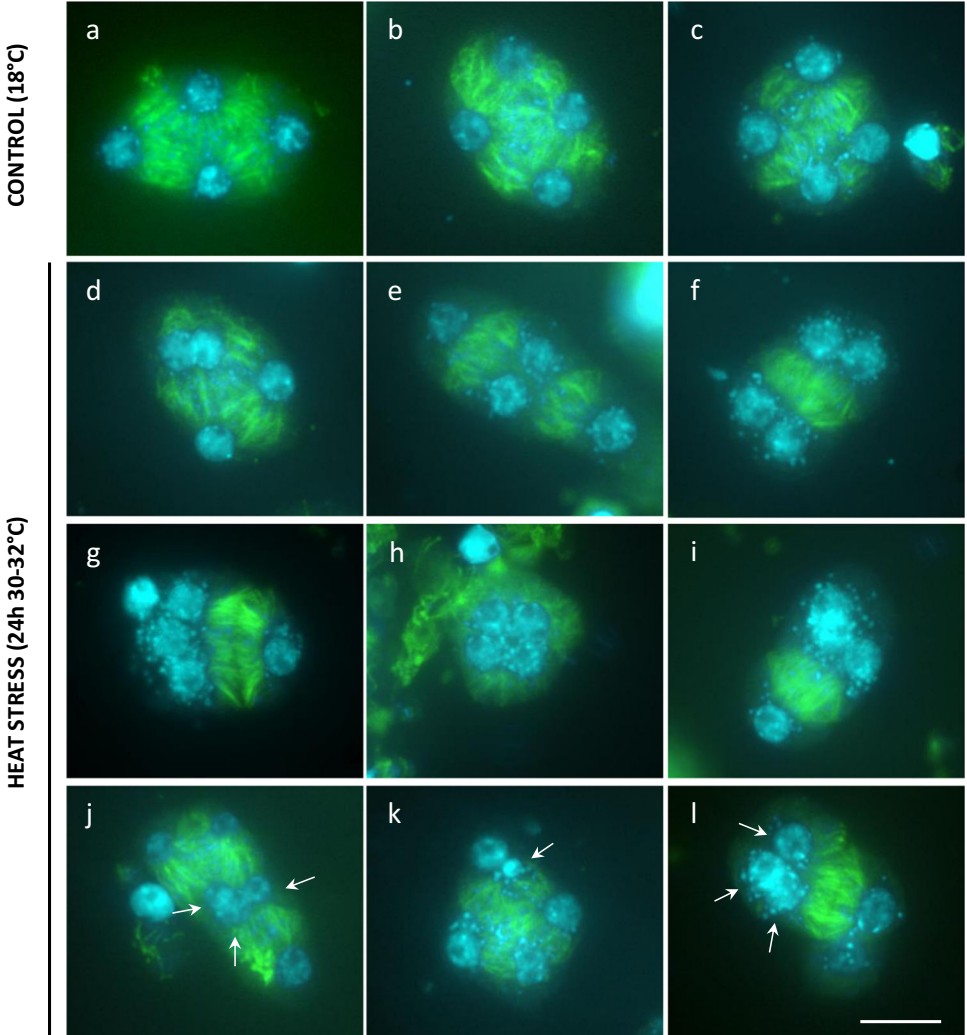

**Fig. 4 Heat-induced defects in MII cell wall formation are attributed to alterations in RMA biogenesis.** Tubulin α immunostaining of *Arabidopsis* Col-0 male meiocytes at telophase II under normal temperatures (18–20 °C, **a**–**c**) and upon heat stress (24 h at 30–32 °C, **d**–**f**). Arrows indicate for the occurrence of two or more nuclei at telophase II that lack the presence of an internuclear phragmoplast-like structure. Scale bar, 10 μm.

fully synapsed chromosomes at pachytene (Figs. 7a and 8a). Next, at diplotene, the synaptomenal complex (SC) degrades to form five highly condensed chromosome pairs (Figs. 7b and 8b) that are physically connected by chiasmata. Metaphase I is marked by the equatorial alignment of all five bivalents (Fig. 8c), and is followed by the segregation of homologous chromosomes to opposite poles by the MI spindle in anaphase I (Figs. 7c and 8d). During the second meiotic cell division, the two haploid chromosome sets align along two perpendicularly oriented metaphase II plates (Fig. 8e), after which the corresponding chromatids segregate to opposite poles. Via the perpendicular orientation of the spindles in MII, male meiosis generates four sets of five chromatids that are spatially arranged in a tetrahedral configuration. Following nucleation and cytokinesis, these four distinct chromatid sets constitute the haploid spore initials in the resulting tetrad (Fig. 8f).

A brief heat-shock treatment (6 h at 30–32 °C) does not alter chromosome appearance and condensation dynamics during the early stages of prophase I. Also, initiation and establishment of synapsis occur similarly as under control conditions. At pachytene, however, heat-stressed PMCs consistently show alterations in SC formation, ranging from short asynaptic DNA loops to more extended chromosomal regions that are not properly synapsed (Fig. 7, $n = 100$). Also, at this stage, a minor subset of heat-stressed PMCs (4%, $n = 100$) exhibits individual chromosomal units that appear isolated in the nucleoplasm (Fig. 7m, asterisk), indicating for putative defects in telomere clustering, chromosome dynamics, or DNA break repair. In later stages, no separated chromosome units or isolated DNA fragments were retrieved (Fig. 7), suggesting that the individual chromosomes or chromosome parts at pachytene do not result from defected DSB repair, but more likely represent chromosomes that do not participate in telomere clustering. At diakinesis, heat-stressed *Arabidopsis* meiocytes consistently reveal five highly condensed bivalents; however, in several PMCs (20%, $n = 20$) interconnecting DNA strands between two or more bivalents were observed (Fig. 7e, h, k, n, arrows), indicating for promiscuous interactions between nonhomologous chromosomes. The equatorial positioning of bivalents in metaphase I and subsequent segregation of chromosomes in anaphase I occurs normal and is not affected by a 6-h heat shock. However, also in anaphase I, a subset of heat-stressed meiocytes (9%: $n = 46$) was found to display ectopic interlinks between segregating chromosomes (Fig. 7, f, i, l, o).

Extended exposure to heat stress (24 h at 30–32 °C) causes similar, albeit more pronounced defects in SC formation and

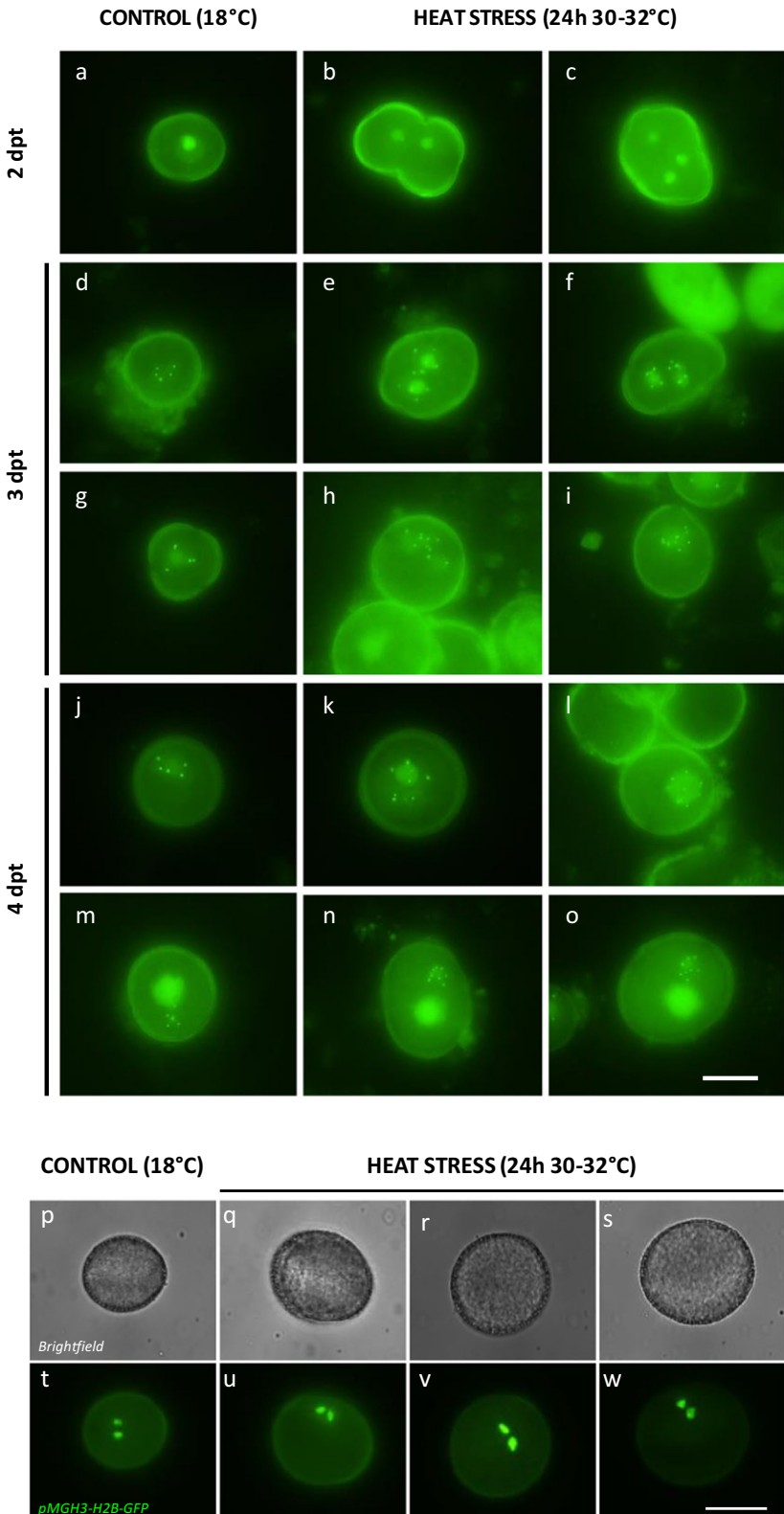

**Fig. 5 Heat-induced bi- and polynuclear spores show nuclear fusion before PMI to yield di- and polyploid pollen grains. a–o** In vivo fluorescent analysis of nuclear dynamics and centromere number during male gametogenesis in spores resulting from *Arabidopsis* male meiocytes that experienced normal temperatures (18 °C, **a**, **d**, **g**, **j**, **m**) and high-temperature stress (24 h at 30–32 °C, **b–c**, **e–f**, **h–i**, **k-l**, **n–o**) using the pWOX2-CENH3-GFP reporter. Scale bar, 10 μm. **p–w** In vivo analysis of the number and size of sperm nuclei in mature pollen originating from control (18 °C, **p**, **t**) and heat-stressed (24 h at 30–32 °C, 7 dpt, **q–s** and **u–w**) *Arabidopsis* male meiocytes using the pMGH3-H2B-GFP reporter. Scale bar, 20 μm.

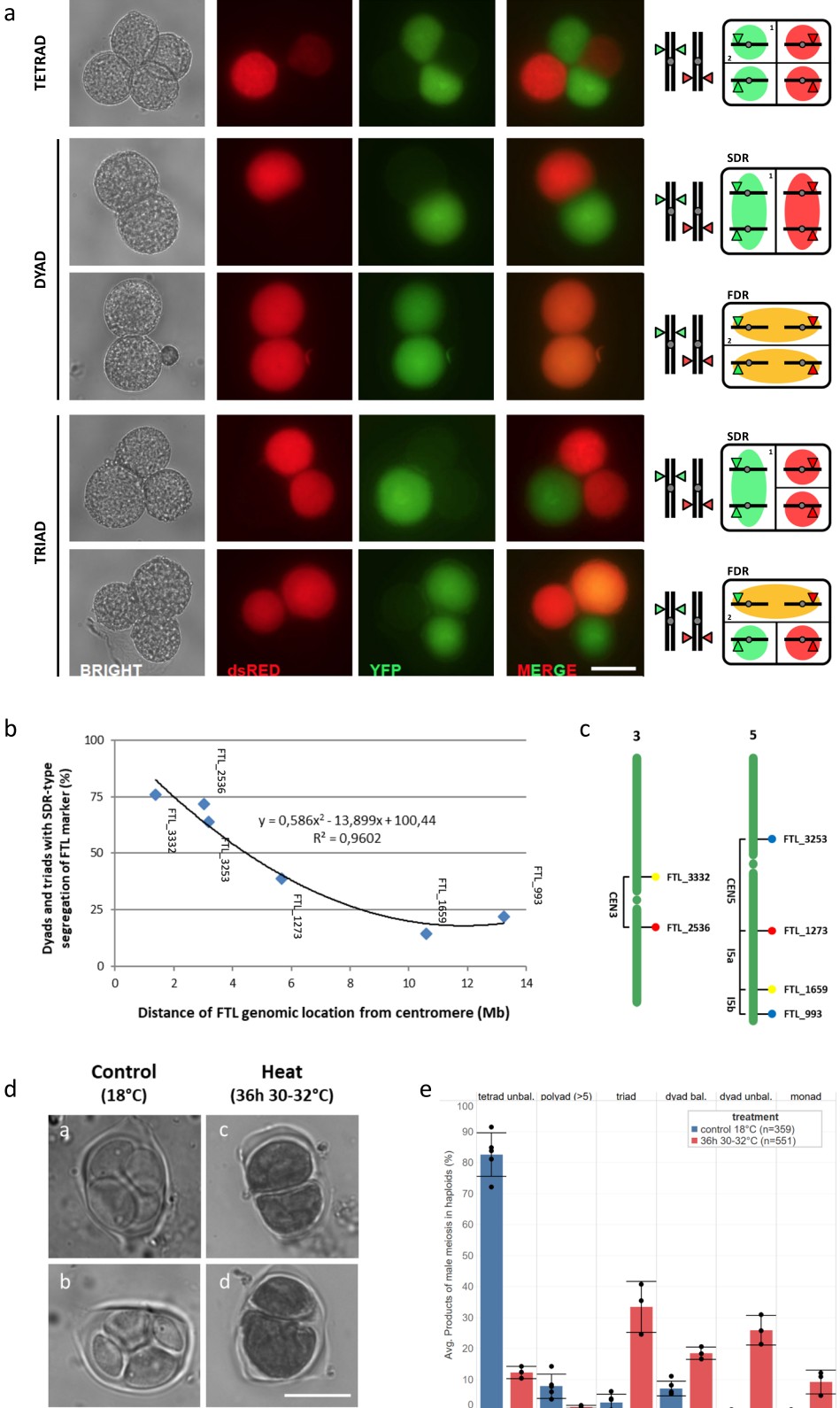

more frequently induces chromosome bridges at anaphase I (16%: $n = 37$) (Fig. 8, j, p, v). In addition, at diakinesis and metaphase I, heat-stressed male meiocytes occasionally display univalent chromosomes (22%, $n = 23$), indicating for partial loss of obligate cross-over formation between homologous partners (Fig. 8h, n, see arrows). These univalents either participate in MI segregation to produce unbalanced chromosome sets (Fig. 8w), or remain

stalled at the cellular midzone to form separate nuclear regions at the start of MII (Fig. 8k, q). Chromatid separation in MII occurs normally under heat. However, in a subset of the resulting tetrad-stage male meiocytes (15%, $n = 33$), two or more nuclei appear clustered and are not separated by a distinct organelle band (Fig. 8l, r). This lack of nuclei separation at the end of MII most likely underpins heat-induced defects in meiotic cell wall

**Fig. 6 Heat-induced meiotic restitution in *Arabidopsis* predominantly yields SDR-type 2n pollen. a** Principle and examples of quantitative analysis of FDR/SDR-type male meiotic restitution in *Arabidopsis qrt1-2$^{-/-}$* plants using segregation analysis of centromere-linked FTL reporters in pollen dyads and triads. Representative bright-field and fluorescent images of *qrt1-2$^{-/-}$* tetrad, dyad, and triad configurations at *Arabidopsis* anthesis displaying segregation of two centromere-linked FTL markers, i.e., FTL_3332 (YFP) and FTL_2536 (dsRed), in the corresponding haploid and diploid pollen grains. Scale bar, 20 μm. **b** Frequency of heat-induced *qrt1-2$^{-/-}$* dyads and triads showing SDR-type segregation of different single FTL reporters in function of their physical distance from the centromere. The closer the FTL reporter is located to the centromere, the higher the frequency of dyads and triads with co-segregation of the corresponding FTL marker, indicating for SDR-type meiotic restitution. Corresponding quantitative data are provided in Supplementary Table 1. **c** Physical position of the different FTL markers on chromosomes 3 and 5 used for FDR/SDR assay together with indications of the corresponding interval. **d** Representative images of tetrad-stage male meiocytes in haploid *Arabidopsis* Col-0 plants under normal conditions (18 °C, **a**, **b**) and upon exposure to heat (24 h at 30–32 °C, **c**, **d**). Scale bar, 10 μm. **e** Comparative analysis of the frequency of different types of male meiotic products formed in haploid *Arabidopsis* plants grown under standard temperature conditions (18 °C) or upon exposure to heat (24 h at 30–32 °C). Statistical differences in the frequency of a specific meiotic product between control conditions and heat stress were tested using one-way ANOVA ($\alpha = 0.05$) and are indicated by asterisks. For each treatment, at least three independent plants were analyzed. The total number of male meiotic products assessed in each treatment is indicated by value *n*.

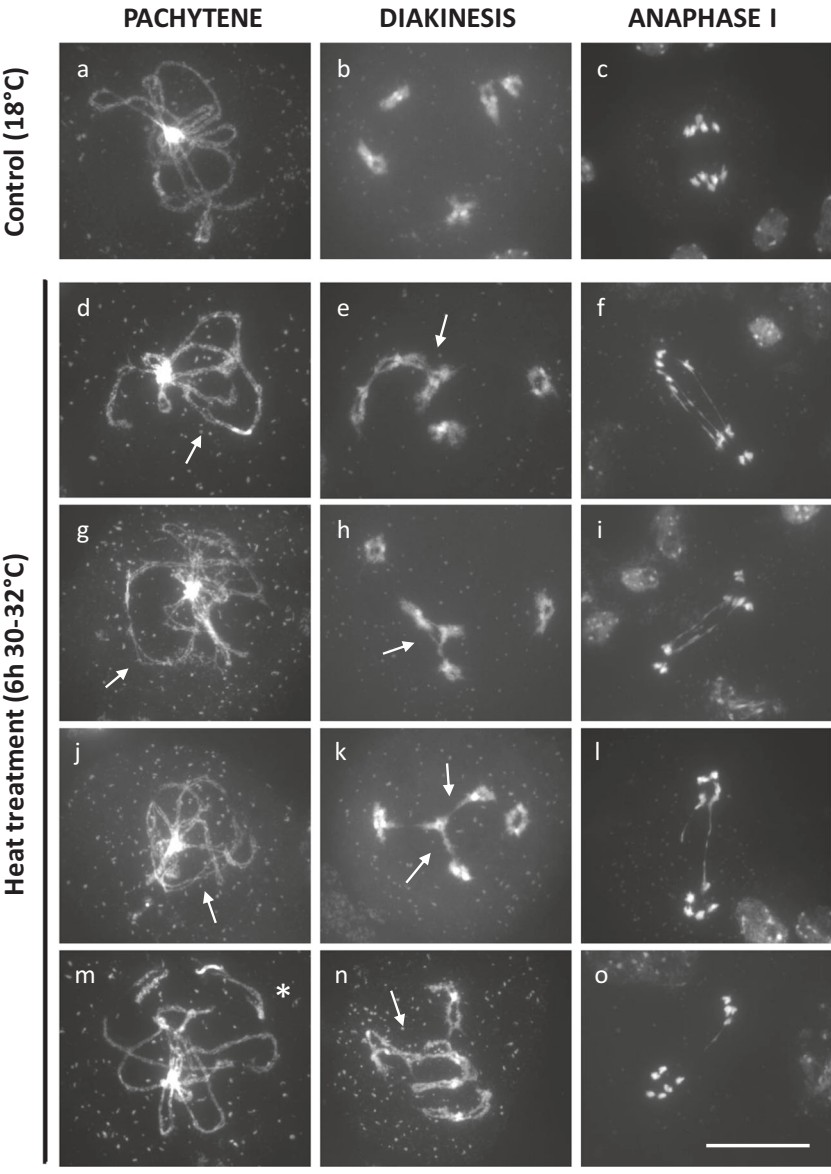

**Fig. 7 Partial asynapsis and anaphase I chromosome bridges in *Arabidopsis* male meiosis exposed to a short period of heat stress (6 h at 30–32 °C).** DAPI-stained chromosome spreads of *Arabidopsis* male meiocytes under standard temperature (18 °C, **a–c**) and upon exposure to a short period of heat stress (6 h, 30–32 °C, **d–o**) at three distinct phases in meiosis I, namely pachytene, diakinesis, and anaphase I. Arrows indicate for chromosomal regions that are not synapsed (**d**, **g**, **j**, **m**) or for ectopic DNA interconnections between nonhomologous chromosomes in diakinesis (**e**, **h**, **k**, **n**). The asterisk indicates for the occurrence of a single chromosome fragment at pachytene. Scale bar, 20 μm.

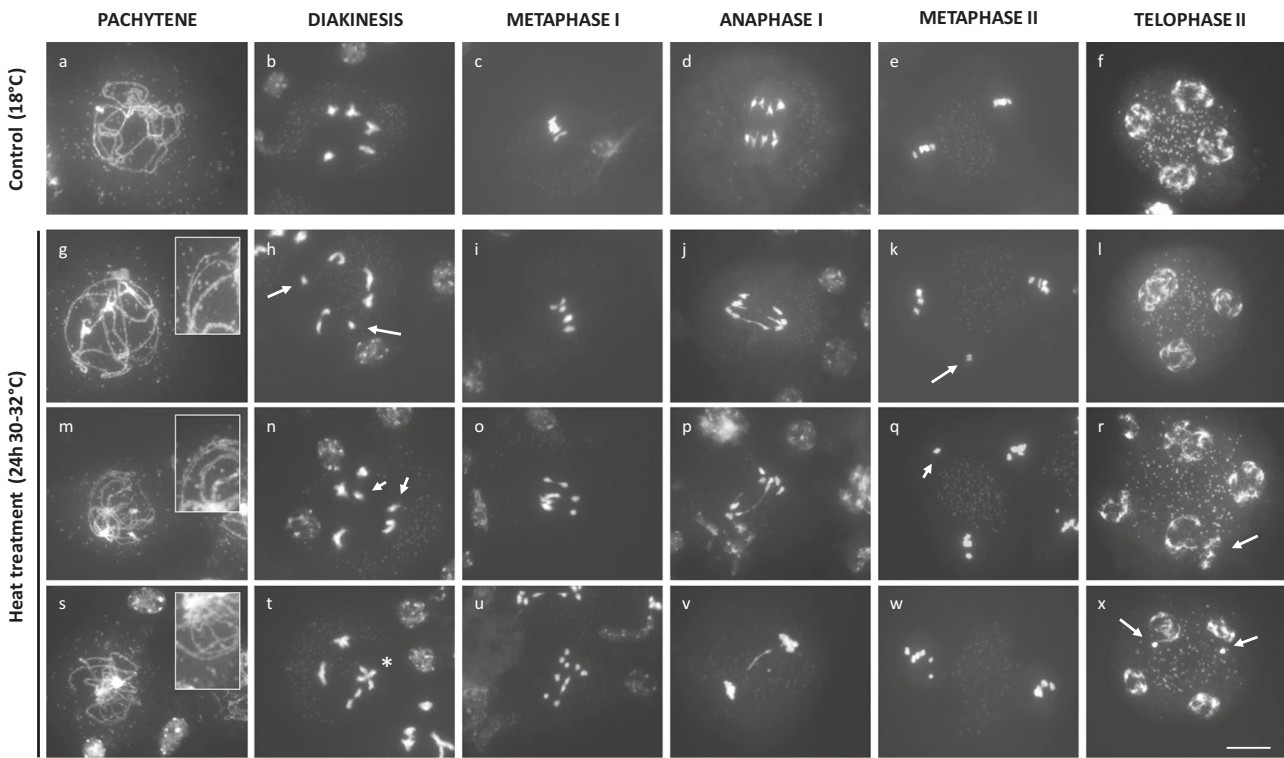

|  | PACHYTENE | DIAKINESIS | METAPHASE I | ANAPHASE I | METAPHASE II | TELOPHASE II |

**Fig. 8 Partial asynapsis, univalents, and anaphase I bridges in *Arabidopsis* male meiosis exposed to a prolonged period of heat stress (24 h at 30–32 °C).** DAPI-stained chromosome spreads of *Arabidopsis* male meiocytes under standard temperature (18 °C, **a–f**) and upon exposure to a prolonged period of heat stress (24 h at 30–32 °C, **g–x**) at different stages in MI and MII. Image insets are magnifications of chromosomal regions that show lack of synapsis (**g**, **m**, **s**). Arrows indicate for the occurrence of univalents in diakinesis (**h**, **n**), aberrant chromosome positioning in metaphase II (**k**, **q**), and occurrence of extra nuclear domains in meiotic tetrads (**r**, **x**). The asterisk indicates ectopic interconnection between bivalent chromosome structures at diakinesis. Scale bar, 10 μm.

formation and forms the basis for bi- or polynuclear microspore formation under heat.

To validate the impact of heat on interhomologous pairing and synapsis in *Arabidopsis thaliana* male meiosis, SC establishment was cytologically monitored via AtASY1 and AtZYP1 immuno-labeling. AtASY1 loads onto the chromatin in early-stage meiotic prophase 1 and associates with the axial/lateral elements of the SC, hence constituting an important structural element in the morphogenesis of the synaptonemal complex. AtZYP1a/b, on the other hand, physically links the axial elements of the interacting chromosomes and thus constitutes a key part of the transverse filament of the synaptonemal complex[19]. Under standard temperature conditions (18–20 °C), both SC proteins exhibit chromosome-specific loading in male meiosis I, in a manner as previously described. AtASY1 initially loads as punctate foci at the chromosomal axial elements in the meiotic interphase, after which the AtASY1 signal extends to cover the whole length of the chromosomes at the end of zygotene (Supplementary Fig. 5). AtZYP1 loads onto paired chromosome regions starting from late leptotene to generate initial domains of homolog synapsis. In subsequent stages, the AtZYP1 signal lengthens along the paired homologs to finally cover the entire length of the chromosomes, i.e., reflecting full synapsis, at pachytene (Supplementary Fig. 5). Under heat stress (24 h, 30–32 °C), no alterations in AtASY1 patterning were observed; AtASY1 exhibits chromosome-specific loading throughout meiotic prophase I in a spatiotemporal manner similar as under control conditions (Supplementary Fig. 5). AtZYP1, on the other hand, shows distinct alterations in prophase I loading or stability under heat. In early- stage meiosis, heat-stressed PMCs reveal focused regions of AtZYP1 loading on paired chromosomes, similar as under control conditions;

however, these never developed into fully extended signals covering the whole chromosome in later prophase I stages (i.e., pachytene). Instead, heat-stressed meiocytes (100%, $n = 18$) consistently revealed distinct chromosomal regions depleted of AtZYP1 and/or domains with only AtZYP1 foci, indicating the local absence of transverse filament deposition and the associated aberrations in SC formation (Supplementary Fig. 5).

**Heat stress alters patterns of male meiotic crossing over in *Arabidopsis thaliana*.** To investigate the overall effect of heat on the frequency and distribution of crossing over in male meiosis, the cross-over rate in eight different genomic intervals was monitored using the pollen FTL bio-assay system (Supplementary Figs. 6 and 7). As FTL-based cross-over analysis is based on the segregation analysis of fluorescent reporters at the mature pollen stage, and it generally takes 6–7 days for a male meiotic end product to develop into a mature pollen grain, FTL segregation analysis was performed in mature pollen that were formed 6, 7, 8, 9, and 10 days following heat stress exposure (24 h at 30–32 °C).

FTL segregation analysis in $qrt1-2^{-/-}$ pollen isolated at 6, 7, 8, and 10 days post heat treatment revealed no substantial shifts in the corresponding meiotic cross-over frequency in all tested genomic intervals compared with control plants that were grown at 18–20 °C. At 9 days post heat treatment, however, pollen FTL segregation analysis showed a strong reduction in cross-over frequency in the centromeric CEN3 interval compared with control plants (Supplementary Fig. 6). As unicellular spores generally take 6–7 days to develop into mature pollen grains and meiosis takes ±33 h, the observed change in CEN3 FTL segregation pattern in pollen assessed at 9 dpt most likely corresponds to shifts in cross-over frequency caused by direct

exposure of male meiocytes to heat. In the other centromeric interval assessed, however, i.e., in CEN5, no distinct shifts in FTL segregation were observed at 9 dpt (Supplementary Fig. 6). This indicates that heat does not alter the meiotic cross-over frequency in all centromeric domains. FTL segregation analysis in $qrt1–2^{-/-}$ pollen isolated at 9 dpt further revealed a distinct, albeit nonsignificant, reduction in cross-over frequency in the I2a interval upon heat (Supplementary Fig. 6). In contrast, in the genomic intervals I1b, I5b, and CEN5, no shifts in meiotic recombination were observed, and in the genomic regions I1c, I2b, and I5a, even slight increases in the cross-over frequency were observed based on FTL segregation in $qrt1–2^{-/-}$ pollen isolated at 9 dpt (Supplementary Table 2). Although shifts in FTL segregation are mostly nonsignificant and show different responses in different genomic regions, they were consistently observed in pollen isolated at 9 days post heat stress, indicating that heat causes slight alterations in the male meiotic cross-over landscape in a transient manner.

**Heat induces nonhomologous recombination in *Arabidopsis* male meiosis.** Anaphase I bridges in heat-stressed male meiosis may result from different types of prophase I alterations, including impaired resolution of cross-over intermediates, faulty resolution of chromosomal interlocks, or ectopic recombination between nonallelic DNA sequences (i.e., nonhomologous recombination).

To determine whether heat-induced chromosome bridges are DSB-dependent, $atspo11–1–3^{-/-}$ plants were incubated for 24 h at 30–32 °C, and MI chromosome dynamics was assayed using DAPI-stained chromosome spreading. Male meiosis in the $atspo11–1–3^{-/-}$ line produces almost no DSBs and hence completely lacks homologous pairing and synapsis, yielding 10 univalents that randomly segregate in MI (100%, $n = 50$). Strikingly, under heat stress, $atspo11–1–3^{-/-}$ PMCs display no ectopic chromosome associations at diakinesis or at anaphase I (Fig. 9, $n = 50$), demonstrating that the occurrence of these erratic chromosome interactions under heat depends on DSB formation and thus likely reflects alterations in DNA break repair or homologous recombination.

To monitor for the occurrence of nonhomologous recombination, we next assessed MI chromosome dynamics in haploid *Arabidopsis* PMCs exposed to heat (24 h at 30–32 °C). Haploid meiocytes only have one copy of the basic chromosome set ($2n = 1× = 5$) and hence show a complete lack of pairing, synapsis, and cross-over formation due to the absence of homologs. In line with this, haploid PMCs consistently reveal five distinct chromosomal units at diakinesis and metaphase I under normal temperatures (Fig. 10, $n = 32$). Contrarily, upon exposure to heat, the majority of haploid PMCs exhibits interchromosomal DNA connections leading to the occurrence of one or more bivalent structures at diakinesis and metaphase I ($n = 17$; 70.5% with one or more bivalents) (Fig. 10). Since haploid meiocytes contain only one copy of the basic chromosome set, heat-induced interchromosomal connections occur between nonallelic regions, and thus indicate for nonhomologous recombination. Hence, anaphase I chromosome bridges occurring in heat-stressed *Arabidopsis* diploid PMCs (Figs. 7 and 8) result from ectopic events of nonhomologous recombination.

## Discussion

Sensitivity of the male reproductive system to heat stress has been reported for various mono- as well as dicotyledonous plants[20]. Our study in *Arabidopsis* shows that high-temperature stress causes alterations in recombination and cytokinesis. Most *Arabidopsis* 2n pollen grains formed under heat stress showed loss of

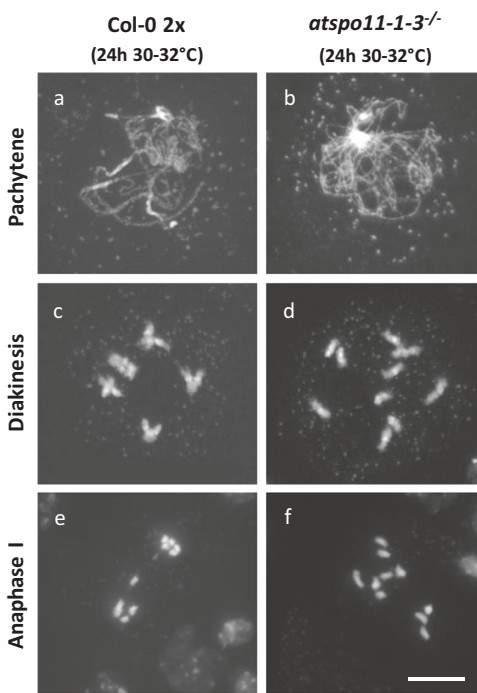

**Fig. 9 Heat-induced anaphase I chromosome bridges are DSB-dependent.** DAPI-stained chromosome spreads of male meiocytes of wild-type (**a**, **c**, **e**) and $atspo11–1–3^{-/-}$ (**b**, **d**, **f**) *Arabidopsis* plants under heat stress (24 h at 30–32 °C) at three distinct MI phases, namely pachytene, diakinesis, and anaphase I. Scale bar, 10 μm.

parental heterozygosity indicating for an SDR-type restitution mechanism. This implies that heat-induced restitution in *Arabidopsis* is specifically caused by defects in the biogenesis of RMA that guide formation of the cell wall between separated chromatids at the end of MII. In rose and poplar, heat interferes with the three-dimensional organization of the syncytial MII spindles, leading to parallel, tripolar, and fused spindle structures and the associated induction of FDR-type restitution[5,10]. Heat stress-induced cytokinesis defects resulted in more triads than dyads, an outcome that is different to what is observed in AtPS1 and Jason mutants where proportionally more dyads occur[21,22]. The AtPS1 and Jason mutants are defective in meiotic II spindle orientation, regularly causing the metaphase II spindle to fuse that generates dyads, whereas spreads of heat-stressed meiocytes never revealed indications for fused spindles. Heat-treated flower buds were fixed right after the treatment and produced tetrads with two proximal nuclei and poorly developed RMA structures between the other two nuclei. In fewer tetrads, three nuclei were adjoined and separated from the fourth by radial microtubules. This mixture of poorly developed microtubule structures and the lack of microtubules between nuclei, suggests that heat stress does not cause the complete disintegration of the RMA, but rather interferes with the initiation or further development of a properly organized microtubule structure in-between nuclei. Poorly developed microtubule structures between nuclei are sufficient to engage in cytokinesis and cell plate formation, as the frequency of monads was extremely low and it may also explain the higher incidence of triads.

But how does heat interfere with meiotic RMA formation and cell wall formation? Studies on mitotic cell division in *Nicotiana tabacum* and the seagrass *Cymodocea nodosa* hint toward the possibility that high-temperature stress changes the organization or stability of microtubules (MT) and thereby deregulates cytoskeletal networks, including the spindle and the

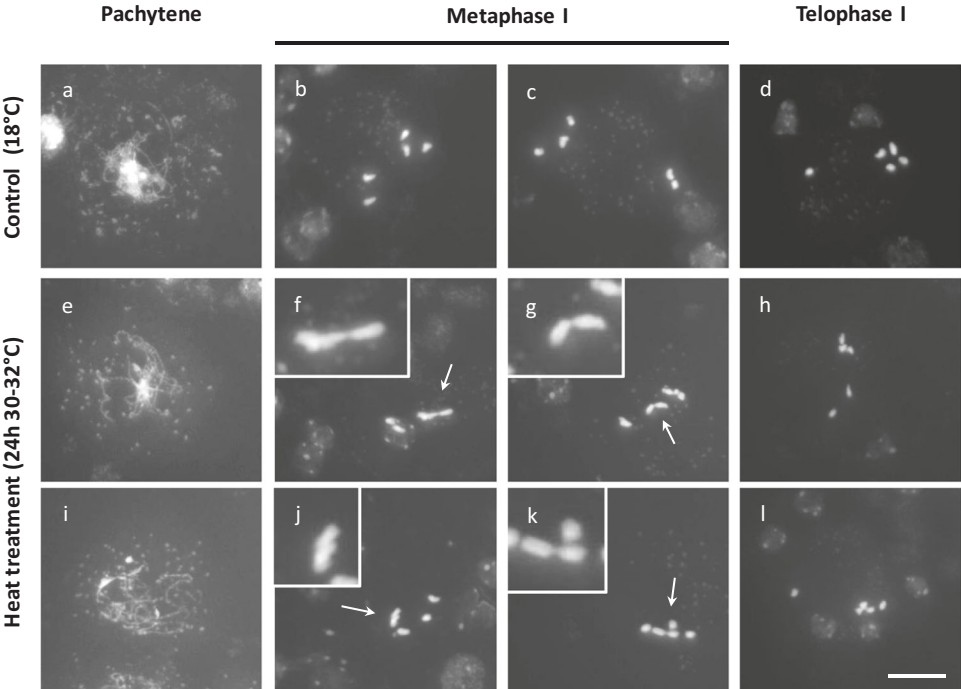

**Fig. 10 Heat stress induces ectopic occurrence of nonhomologous recombination in male meiosis of haploid *Arabidopsis*.** DAPI-stained chromosome spreads of male meiocytes of wild-type haploid *Arabidopsis* Col-0 plants under standard temperature conditions (18–20 °C, **a–d**) and upon heat stress (24 h at 30–32 °C, **e–l**). Arrows indicate for ectopic physical connections between nonhomologous chromosomes in metaphase I. Scale bar,10 μm.

phragmoplast. As a result of this MT instability, heat causes mitotic arrest at telophase and the absence of cytokinesis, leading to defected cell wall formation[23,24]. Studies on male meiosis in several plant species revealed a similar sensitivity of MT structures to high-temperature stress. In poplar, heat stress interferes with the biogenesis of all types of meiotic MT arrays, including MI/MII spindles and RMAs, with restoration of cytoskeletal figures eventually causing induction of meiotic restitution[10]. Strikingly, contrary to mammals, heat-induced alterations in meiotic MT figures in plants do not lead to an arrest in MI or MII cell division, indicating for the absence of a stringent checkpoint mechanism controlling meiotic cell division in plants. In line with this, the spindle assembly checkpoint (SAC) in plants has been found to be less stringent under stress, allowing re-initialization of cell cycle progression upon return to normal conditions[25]. This is further corroborated by several mutants, such as the *Arabidopsis atk1* and the maize *multipolar spindle* (*mps*) and *divergent spindle-1* (*dv1*) mutant, that all show defective MT dynamics in meiosis, but still complete the meiotic cell division program[26–29].

At 30–32 °C, *Arabidopsis thaliana* male meiosis exhibits alterations in chiasma formation and chromosome segregation, leading to polyads that harbor aneuploid spores[30,31]. Heat-stressed *Arabidopsis* PMCs exhibit reduced cross-over frequency within specific genomic regions, such as the centromeric interval on chromosome 3, and occasionally show univalents at MI, indicating for loss of obligate cross-over formation. In other genomic regions, however, the meiotic cross-over rate appears unaffected or even increased, indicating that heat does not merely affect the overall genome-wide cross-over level, but instead causes a spatial redistribution of cross-over events along the chromosome. The overall effect of varying temperatures on meiotic cross-over rate and distribution in *Arabidopsis* was recently investigated by several groups[16,18,32]. These studies revealed that heat increases the overall recombination rate in male meiosis, and that the extra crossovers under heat are formed via the class I interference-sensitive pathway. Similarly, in barley, the male

meiotic cross-over landscape is temperature-dependent, with moderate heat stress (25–30 °C) increasing the overall cross-over rate and shifting cross-over distribution from the distal telomeric regions to more interstitially located domains[16]. This heat-induced shift in spatial cross-over designation is caused by weakening of the otherwise strong spatiotemporal polarization of initiation of homologous pairing and recombination along the chromosome[15]. How the distribution of meiotic cross-over formation is influenced by temperature remains to be discovered.

In eukaryotes, meiotic cross-over designation is strictly regulated at the molecular level, with several integrated regulatory mechanisms determining the meiotic cross-over landscape. This inherent control of cross-over distribution is highly sensitive to temperature. A question remains, however, which regulatory element(s) underpinning spatial cross-over designation are affected? In barley, heat-induced changes in cross-over patterning are positively correlated with the length of synaptonemal complexes at pachytene[16]. Our study in *Arabidopsis* here demonstrates that heat influences the process of synapsis, but instead of increasing SC length, temperatures of 30–32 °C were found to interfere with the biogenesis or stability of the SC. Immunostaining of the major axial and transverse SC constituents revealed that heat stress does not affect AtASY1 dynamics, indicating stable chromosome axis configuration in prophase I. AtZYP1, on the other hand, exhibited a patched localization pattern in pachytene-stage PMCs. This demonstrates that SC buildup, and particularly biogenesis of the transverse element, during early-stage prophase I is highly sensitive to high-temperature stress. A similar sensitivity of the meiotic SC structure and the transverse element to heat has also been shown to occur in several other eukaryotic organisms, including *Caenorhabditis elegans*[33] and *Schistocerca gregaria*[34]. However, despite this common response, it is yet unclear whether heat-induced defects in SC result from a direct physical disruption of structural SC components, or instead indirectly originate from aberrations in upstream regulatory processes that are crucial for SC biogenesis. Based on the

observation of thickenings in the chromosome axis of heat-stressed meiocytes in several plants and other organisms, it has been postulated that heat directly interferes with SC formation[15,33,35,36]. However, cytological analyses in *Arabidopsis* did not reveal SC polycomplexes or axis thickenings in heat-stressed PMCs, suggesting that heat affects SC biogenesis in an indirect manner.

In the current context of global warming and climate change, our findings here provide relevant insights into the genomic stability of male reproductive development under conditions of heat stress. With local climates showing more temperature extremes, plants may produce higher levels of di- and aneuploid gametes, thereby increasing the odds of creating novel karyotypes, such as poly- and aneuploids. Evolutionary studies in plants and other organisms have shown that newly formed polyploids often constitute a dead end[37,38], whereas other chromosomal variations, such as aneuploidy and dysploidy, are more persistent and often lead to the establishment of new genotypes, particularly under environmentally adverse conditions[39–41]. In this framework, our data provide a possible cellular mechanism(s) by which novel plant karyotypes may emerge. Temperature sensitivity of male meiosis, and particularly cross-over formation and MII cytokinesis, constitutes a putative mechanism for karyotype evolution in plants, and hence forms a valid path for adaptive genome evolution under environmentally challenging conditions.

## Methods

**Plant material and growth conditions**. The Arabidopsis Columbia-0 (Col-0) wild type was obtained from the Nottingham Arabidopsis Stock Centre (NASC). The $qrt1-2^{-/-}$ line (N8846) was donated by G. Copenhaver, and segregating progeny seed resulting from the selfing of heterozygous $atspo11-1-3^{+/-}$ (SALK_146172) and $atps1-1^{+/-}$ (SALK_078818) parental lines was kindly provided by R. Mercier[21]. For assaying nuclear dynamics and chromosome quantification during male gametogenesis, *Arabidopsis* plants harboring *pMGH3::H2B-GFP* or *pWOX2::CENH3-GFP* constructs were used, respectively[42,43]. The *Arabidopsis* FTL reporter lines, used for genotypic analysis of unreduced 2n gametes (FDR/SDR assay) and quantitative analysis of male meiotic CO frequency, were kindly donated by G. Copenhaver[44]. Except for CEN3, CEN5, and I5ab, all these FTL lines are in the $qrt1-2^{-/-}$ background and are hemizygous for enclosed fluorescent reporter constructs. Single reporter lines homozygous for the FTL3332 (YFP) and FTL2536 (DsRed2) fluorescent reporter were intercrossed to generate the CEN3 line. Similarly, plants homozygous for the FTL1273 (DsRed2) and FTL3253 (AmCyan) fluorescent reporter construct were intercrossed to develop the CEN5 line. The resulting CEN3 and CEN5 lines are thus hemizygous for two different FTL reporters and cover the centromeric region of chromosomes 3 and 5, respectively. Also, the homozygous FTL I5ab $qrt1-2^{-/-}$ line was hybridized with $qrt1-2^{-/-}$ plants that lack any fluorescent reporter to generate F1 plants that are hemizygous for all three I5ab FTL reporter constructs and that are homozygous for $qrt1-2^{-/-}$ (i.e., production of pollen tetrads).

Haploid *Arabidopsis* plants were generated by intercrossing diploid Col-0 plants as pollen donor with the seedGFP-HI (Haploid Inducer) line as pollen acceptor and by subsequent analysis of the resulting F1 plants[45]. In brief, F1 plants were first phenotypically selected based on the absence of filled seed pods (haploid *Arabidopsis* plants are sterile), after which the haploid somatic ploidy level was validated using DNA flow cytometry.

*Arabidopsis* seeds were surface-sterilized using chloric gas (±4 h) and sown in vitro on K1 medium. Following stratification (3 days at 4–5 °C in the dark), plates were transferred to standard growth conditions (i.e., 12-h light/12-h dark; 18–20 °C) to induce in vitro seed germination. Young 6–8-day-old seedlings were transferred to soil and cultivated in growth chambers at an ambient temperature of 18–20 °C, photoperiod of 12-h light/12-h dark, and <70% humidity. After 4 weeks of vegetative growth, photoperiod was changed to a 16-h light/8-h dark regime to stimulate flowering.

For the heat-shock treatments, healthy and green flowering plants were transferred to a climate chamber with a set ambient temperature of 26–28 °C or 30–32 °C for different durations (6 h, 12 h, 24 h, 36 h, and 48 h) depending on the specific experimental assay. During heat treatment, other growth parameters were maintained stable: with a photoperiod of 16-h light/8-h dark and <70% humidity. Following heat treatment, *Arabidopsis* plants were immediately returned to standard growth conditions to minimize the adverse effects on plant growth and development. Depending on the specific experimental setup, flower buds containing developing PMCs, microspores, or mature pollen grains were isolated, either directly following heat stress treatment or after 1–10 days following heat stress exposure.

**Histology and cytology of male sporogenesis**. Cytology of nuclear dynamics and cell division in male meiosis and early-stage microgametogenesis was performed by lactopropionic orcein staining[46]. Callose cell wall staining of male meiocytes (tetrad stage) was performed by squashing stage 9 flower buds in a drop of aniline blue solution (0.1% [m/v] in 0.033% $K_3PO_4$ [m/v]) and subsequent visualization using fluorescent microscopy. Chromosome dynamics and behavior in *Arabidopsis* male meiocytes were visualized using the standard chromosome-spreading method[47] with minor modifications[22]. Briefly, following fixation of meiotic buds in 3:1 ethanol:acetic acid (minimum of 24-h fixation at 4–5 °C), buds were rinsed in distilled water and in 10 mM citrate buffer at pH 4.5, and incubated in an enzyme mixture of 0.3% (w/v) pectolyase (Sigma) and 0.3% (w/v) cellulase (Sigma) in 10 mM citrate buffer at 37 °C in a moisture chamber for 1.5 h. Digested buds were rinsed and stored at 4 °C in citrate buffer. A single enzyme-digested bud was then transferred to a slide, macerated with a needle in a small drop of 60% acetic acid, and stirred gently on a hotplate at 45 °C for 30 s. The air-dried slide was then flooded with copious amounts of freshly made 3:1 ethanol:acetic acid and left to dry at room temperature. Finally, slides were stained by adding 25 µL of 4′,6-diamidino-2-phenylindole (DAPI: 1 µg mL$^{-1}$) in Vectashield medium (Vector Laboratories), mounted with a coverslip, and squashed between filter paper to remove excess liquid.

Microtubule (MT) structures and cytological figures in *Arabidopsis* male meiosis, such as MII spindles and telophase II RMAs, were visualized using the α-tubulin immunolocalization assay[22]. Tubulin immunodetection was performed using a rat α-tubulin primary antibody (0.3% [v/v], clone B-5-1-2; Sigma-Aldrich) and a FITC-labeled goat anti-rat IgG secondary antibody (0.5% [v/v], ab6840, Abcam).

**Genetic analysis of meiotic restitution and cross-over frequency using FTL markers**. FDR–SDR analysis of unreduced pollen grains in heat-stressed *Arabidopsis* male meiocytes was performed using three FTL reporter combinations: CEN3, CEN5, and I5ab (Supplementary Fig. 7). Diploid pollen was genetically characterized at maturity by monitoring the segregation of hemizygous FTL fluorescent reporters, closely linked to the centromere, in restituted dyads and triads of *Arabidopsis* $qrt1-2^{-/-}$ plants. Upon heat-shock exposure (24 h at 30–32 °C), mature flowers of $qrt1-2^{-/-}$ plants hemizygous for each of the three FTL reporter combinations were isolated 6–8 dpt, and segregation of the fluorescent signal in $qrt1-2^{-/-}$ pollen dyads and triads was assayed using fluorescence microscopy. In this assay, full fluorescent dyads (i.e., both 2n pollen grains in the dyad are fluorescent) indicate for a disjoining of sister chromatids and thus for FDR-type restitution (Fig. 6a). Alternatively, $qrt1-2^{-/-}$ pollen dyads with one fluorescent and one nonfluorescent spore suggest for co-segregation of sister chromatids with the FTL reporter, hence indicating for SDR-type restitution (Fig. 6a). Similarly, triads in which the diploid 2n pollen grain together with one of the two single-haploid pollen grains exhibits expression of the centromere-linked FTL reporter indicate for FDR-type restitution, whereas triads that show fluorescent expression either in the 2n pollen grain or in the two smaller-sized haploid pollen grains reflect SDR-type restitution (Fig. 6a).

Quantification of the male meiotic recombination frequency in specific genomic intervals was performed using the pollen-specific FTL reporter system, as previously outlined[48]. In brief, the FTL system for cross-over quantification takes advantage of readily available *Arabidopsis* $qrt1-2^{-/-}$ FTL reporter lines that are hemizygous for two or more genetically linked recombinant constructs that each express a different fluorochrome (i.e., eCFP, YFP, and dsRed) in the cytoplasm of mature pollen grains. By assessing the segregation of the corresponding fluorochromes in the resulting $qrt1-2^{-/-}$ pollen tetrads, this system allows for the easy and straightforward quantification of the meiotic cross-over frequency in the corresponding genomic interval. It is important to note that readout of the cross-over frequency in specific meiocytes using the FTL system can only be performed later in development, i.e., when the resulting spores have developed into mature pollen grains. Because of that, putative effects of a heat-shock treatment (24 h at 30–32 °C) on meiotic recombination were here assessed by visually monitoring FTL segregation in mature pollen tetrads that were released (i.e., at anthesis) in the days subsequent to heat stress treatment. As it generally takes 6–7 days for a meiotically formed uninuclear spore to develop into a mature pollen grain in *Arabidopsis*, putative effects of a 1-day heat-shock exposure (30–32 °C) on meiotic cross-over frequency were quantified by assaying FTL segregation in the resulting $qrt1-2^{-/-}$ pollen tetrads from 6 to 10 days following heat treatment.

**Microscopy**. Both bright-field and (epi-)fluorescence microscopy were performed using an Olympus IX81 inverted fluorescence microscope equipped with an X-Cite 120 LED Boost lamp. Images were captured using an Olympus XM10 camera. Bifluorescent images and Z stacks were processed using ImageJ.

**Statistics and reproducibility**. Quantitative analysis of tetrad-stage meiotic end products, microspore, and mature pollen configurations in *Arabidopsis thaliana* wild-type, mutant, and FTL reporter lines under normal temperature (18–20 °C) as well as the following various heat treatments were performed immediately after the treatment, unless otherwise stated, and consistently performed for a minimum number of three independent plants (i.e., biological repeats). Immature flower buds

or flowers at anthesis were thereby always isolated from the primary stem to avoid putative alterations caused by developmental variation. For each experimental assay included in this study, the exact number of meiocytes, spores, or restituted pollen figures is recorded and listed in the corresponding table or figure (see value *n*).

Statistical analysis of data and comparison of means was executed using the SPSS v25 software. For each quantitative analysis with a minimum of three repeats, the resulting data were checked for normal distribution using the Shapiro–Wilk test ($\alpha = 0.05$). For data that were not normally distributed, statistical comparison of medians was performed using the nonparametric Kruskal–Wallis test. For normally distributed data, homogeneity of variances was assessed using Levene's test, and statistical comparison of the means was performed via t test or via one-way ANOVA with multi-comparison Tukey's HSD post hoc test. Significant differences between different treatments, mutants, and/or lines are indicated in the corresponding figures using a star or a letter.

**Reporting summary**. Further information on research design is available in the Nature Research Reporting Summary linked to this article.

## Data availability
All data generated or analyzed during this study are included in this published article (and its supplementary information files).

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

## Acknowledgements
We thank Chris Franklin for providing an antibody to detect Asy1 and Zyp1.

## Author contributions
N.D.S. and D.G. developed the research concept, N.D.S. executed the experiments, and N.D.S. and D.G. interpreted the results, compiled, and wrote the paper.

## Competing interests
The authors declare no competing interests.
