## [Peer Review File · Communications Biology]

Reviewers' comments:

Reviewer #1 (Remarks to the Author):

In this manuscript, De Storme and Geelen evaluate the consequences of heat stress on *Arabidopsis thaliana* male meiocytes. They demonstrate that elevated temperatures produce alterations in the recombination rate in different genomic intervals and deficiencies in chiasma formation, as well as restitution of meiosis by defects in cell wall formation during the second division. The results derived from this work are potentially interesting, because they allow a better understanding of the consequences of an increase in temperature on plant fertility. They also offer a possible explanation for the origin of diploid gametes that can give rise to polyploidy. However, in my opinion I consider that there are a number of issues in this study that require further analyses. First of all there is a lack of uniformity in the heat treatments applied in the various experiments (different temperatures and different time intervals). I understand that on some occasions the conditions cannot be standardized due to the specific characteristics of each analysis (in fact sometimes a normal temperature recovery period is applied after the heat treatment), but no justification is given either. For example, MI univalents and anaphase I bridges are observed in meiocytes exposed to 24h at 30-32°C, whereas crossover frequency is measured at 6, 7, 8 and 10 days after heat treatment (26h at 30-32°C).

Secondly, a proper quantification is absolutely necessary. In most experiments no information is given on the number of cells analysed, nor on the number of plants. Statistical analyses are also missing. On the other hand, although the results of previous studies related to temperature and recombination are commented on, the comparison with the results obtained is not contextualised. In these previous works (Modliszewski et al., 2018; Lloyd et al., 2018), a hyper-recombinant phenotype is revealed as a consequence of thermal stress, due to an increase of class I (interference sensitive) COs. CO assurance is dependent on this class of COs and in this study the authors have observed loss of obligate CO formation ("heat-stressed meiocytes OFTEN display univalent chromosomes"). Nevertheless, in the discussion the authors mention that their findings concur with these previous results. A possible explanation for the discrepancy is that higher temperatures (up to 32°C) are applied in this work compared to the cited works (28°C). Lloyd et al. (2018) indicate that meiotic recombination has a U-shaped response to temperature in *Arabidopsis*. According to the results shown in the present work, the scenario seems to be a little more complicated, but this issue is not discussed in depth.

Finally, the authors state that the phenotypes observed suggest the existence of a "regulatory mechanism that stochastically impacts CO homeostasis". Homeostasis is a term that is also included in the title of this manuscript, despite it is not defined throughout the text. This CO buffering mechanism ensures the maintenance of the CO number (at the expense of non-crossovers) when DSBs are reduced. In this work, neither the number of DSBs nor the total number of COs are evaluated (only recombination is analyzed in some genomic intervals), hence it cannot be indicated that heat produces alterations in the homeostatic regulation of COs.

More specific issues:

- Line 94 (introduction): "Heat-induced shifts in CO rate and distribution significantly correlate with structural alterations in synaptonemal complex (SC) length, indicating for a putative mechanistic control via SC dynamics (Phillips et al., 2015; Lloyd et al., 2018)". However, in Lloyd et al. (2018): "In contrast to barley, we observed that in *A. thaliana* total SC length significantly decreased with increasing temperatures", and "the increase in crossover number at elevated temperatures cannot be explained by an increase in SC length".

- Meiosis lasts 33 hours at 18.5°C (Armstrong et al., 2003. Sex. Plant Repr. 16:141-9). The heat can produce a variation of this time, most likely a delay. The analyses conducted at 24h-48h at the latest (e.g. lines 115, 158, 238, 250) seems insufficient to make a proper assessment of the consequences of heat treatment. Variable results may be obtained depending on the stage in which the meiocytes are when heat treatment is applied.

- Line 260 (results): assertions in relation to synapsis are not accompanied by quantification. In addition, to conclude the existence of a deficiency in the formation of the synaptonemal complex, DAPI staining is insufficient. It is necessary to perform an immunolocalisation to detect both ASY1 and ZYP1 proteins.

- Line 267 (results): What do you mean with "individual chromosome units"? the frequency is not indicated ("a minor subset"). Apparently in the picture (Fig. 5m) the "unit" seems to be a fragment. It must be taken into account that heat treatment can affect chromatin and it can be more easily

fragmented after the spreading protocol. These "fragments" may not be a direct consequence of the heat effect. Regarding the sentence "in later stages, no indications of chromosome fragmentation were retrieved" a clear fragment appears in Fig. 6v (telophase I).

- The putative "promiscuous interactions" (line 274) between different chromosomes have to be confirmed by FISH, to ensure that they take place between non-homologous chromosomes.
- The analysis of the CO frequency at different genomic intervals were made at 6, 7, 8 and 10 dpt. This means that the meiocytes analyzed started meiosis after heat treatment. This situation should be considered in the interpretation of the results.
- Line 410 (discussion): "Our findings therefore support the view that heat-induced alterations in CO homeostasis are caused by a desynchronization of CO establishment and meiotic cell cycle progression." Since no analysis of the duration of meiosis is carried out this cannot be assumed.
- Line 425 (discussion): "it is conceivable that heat induced shifts in CO patterning are not directly inferred by regulatory elements acting in trans, but instead are caused by cis factors such as altered chromatin status or DNA methylation patterning". In the discussion there is a whole paragraph devoted to a possible effect on epigenetic marks. Although it might be a possibility in this study no experiment has been conducted to check for possible epigenetic changes. In my opinion, this part of the discussion should be reduced.

Minor comments:

- Please include the definition of SDR (Second Division Restitution) in the abstract.
- The word "ectopic" appears several times throughout the text. Sometimes it refers to the recombination between non-homologous sequences, but other times it does not have a clear meaning (e.g. lines 46, 207, 235).
- Please standardize the nomenclature for the mutants (e.g. *atspo11-1-3* / *sun1sun2*).

Reviewer #2 (Remarks to the Author):

The manuscript written by Nico and Geelen comprehensively demonstrates how temperature affects male meiotic restitution in Arabidopsis by convincing cytological and genetic approaches. First, they found that heat stress causes restitution of Arabidopsis male meiosis mainly due to the defects of cell wall formation in Meiosis II. Consistent observations of di- and polyploids from microspore early stage to mature pollen evidenced that heat-induced male meiotic restitution leads to higher ploidy gametes. In order to identify the cause, they experimented on centromere-linked FTL lines and Col-0 haploid plants to reveal that restitution primarily generates from SDR. In addition, they reported the cytological phenotype of heat-stressed meiocytes, which corresponded with CO alteration eventually, and inferred the occurrence of non-homologous recombination induced by heat.

This is a very nice piece of work with concise and well-organized results, and clear, straight-forward figures. However, authors should pay more attention to the details. There are more typos and mistakes in this manuscript than usual.

Reviewer #1 (Remarks to the Author):

In this manuscript, De Storme and Geelen evaluate the consequences of heat stress on *Arabidopsis thaliana* male meiocytes. They demonstrate that elevated temperatures produce alterations in the recombination rate in different genomic intervals and deficiencies in chiasma formation, as well as restitution of meiosis by defects in cell wall formation during the second division. The results derived from this work are potentially interesting, because they allow a better understanding of the consequences of an increase in temperature on plant fertility. They also offer a possible explanation for the origin of diploid gametes that can give rise to polyploidy. However, in my opinion I consider that there are a number of issues in this study that require further analyses.

First of all there is a lack of uniformity in the heat treatments applied in the various experiments (different temperatures and different time intervals). I understand that on some occasions the conditions cannot be standardized due to the specific characteristics of each analysis (in fact sometimes a normal temperature recovery period is applied after the heat treatment), but no justification is given either. For example, MI univalents and anaphase I bridges are observed in meiocytes exposed to 24h at 30-32°C, whereas crossover frequency is measured at 6, 7, 8 and 10 days after heat treatment (26h at 30-32°C).

The majority of experiments were performed in a standardized way applying heat stress at 12h, 24h or 36h. Because it often took several hours to analyse all biological samples, the feasibility required varying heat treatment periods, such as 26h or 40h. We preferred to analyse multiple plants in a single experiment to minimize other factors such as the plant history etc. In practice, the experiments were performed in a highly uniform way, and because of that we have – for most experiment with seemingly deviating treatment periods – adjusted the described length of the heat shock treatment to the standard duration times of 12h, 24h and 36h.

In some assays and in case of specific research questions, we focussed on specific developmental stages or on specific durations of heat stress. For example, in specific experiments for 2n pollen FDR/SDR analysis and FTL-based determination of shifts in CO frequency and distribution, the methodology requires one to assess the mature pollen grains 7-10 days following heat stress treatment, as these are the spores that directly result from the meiocytes that have been exposed to the temperature shock. FTL-based analysis of heat-induced SDR/FDR restitution or shifts in CO distribution cannot be performed by analysing pollen at the same day of the heat treatment, since these result from PMCs that have undergone meiosis 7-10 days before the heat shock treatment. In cases where we have opted to use adjusted heat treatment conditions, i.e. in order to answer a specific research question, we have included a justification in the results section of the manuscript.

Secondly, a proper quantification is absolutely necessary. In most experiments no information is given on the number of cells analysed, nor on the number of plants. Statistical analyses are also missing.

The details on the quantification of number of samples, datapoints, biological repeats, number of cells are now systematically included. This refers to the tetrad analyses, FDR/SDR, recombination frequencies, etc. For all the comparative results, we have included the statistical analysis to assess for putative significant differences between mean value of different treatments. For this, we used the SPSS version 25 software to first validate the normal distribution of obtained data populations after which

statistical analyses were either performed using t-test, one-way ANOVA or non-parametric tests, such as the Kruskal-Wallis test. The type of statistic used is included in the figure legends.

On the other hand, although the results of previous studies related to temperature and recombination are commented on, the comparison with the results obtained is not contextualised. In these previous works (Modliszewski et al., 2018; Lloyd et al., 2018), a hyper-recombinant phenotype is revealed as a consequence of thermal stress, due to an increase of class I (interference sensitive) COs. CO assurance is dependent on this class of COs and in this study the authors have observed loss of obligate CO formation (“heat-stressed meiocytes OFTEN display univalent chromosomes”). Nevertheless, in the discussion the authors mention that their findings concur with these previous results. A possible explanation for the discrepancy is that higher temperatures (up to 32°C) are applied in this work compared to the cited works (28°C). Lloyd et al. (2018) indicate that meiotic recombination has a U-shaped response to temperature in Arabidopsis. According to the results shown in the present work, the scenario seems to be a little more complicated, but this issue is not discussed in depth.

We have included an additional paragraph in the discussion section to position our results in the context of the earlier reports that may have given an impartial impression on the impact of heat stress on CO frequency and distribution. We speculate that at higher temperature there may be other factors involved in CO formation that influence CO designation. In our opinion, a different approach will be needed to obtain a much higher numbers of datapoints to determine a heat sensitivity map of the entire genome. This is however beyond the scope of this paper.

Finally, the authors state that the phenotypes observed suggest the existence of a “regulatory mechanism that stochastically impacts CO homeostasis”. Homeostasis is a term that is also included in the title of this manuscript, despite it is not defined throughout the text. This CO buffering mechanism ensures the maintenance of the CO number (at the expense of non-crossovers) when DSBs are reduced. In this work, neither the number of DSBs nor the total number of COs are evaluated (only recombination is analyzed in some genomic intervals), hence it cannot be indicated that heat produces alterations in the homeostatic regulation of COs.

We fully agree with this remark. In the first version of our manuscript the term ‘CO homeostasis’ was used to refer to the complete set of mechanisms that regulate both the frequency and spatial distribution of COs along the Arabidopsis genome (i.e. the CO landscape). However, this is not correct, since, as also indicated by reviewer #1, the term ‘CO homeostasis’ specifically refers to the buffering mechanism that guarantees a stable number of COs under varying levels of double strand breaks (DSBs). We do not assess the relative level of COs in relation to the DSB frequency, and because of that we cannot make claims about the impact of heat stress on CO homeostasis. As this was not the intention of our research study, and we thus actually used the term ‘CO homeostasis’ in the wrong context, we have adjusted the revised manuscript by replacing the term ‘CO homeostasis’ by the terms ‘CO distribution’ or ‘CO designation’.

More specific issues:

Line 94 (introduction): “Heat-induced shifts in CO rate and distribution significantly correlate with structural alterations in synaptonemal complex (SC) length, indicating for a putative mechanistic control via SC dynamics (Phillips et al., 2015; Lloyd et al., 2018)”. However, in Lloyd et al. (2018): “In contrast to barley, we observed that in *A. thaliana* total SC length significantly decreased with increasing temperatures”, and “the increase in crossover number at elevated temperatures cannot be explained by an increase in SC length”.

Meiosis lasts 33 hours at 18.5°C (Armstrong et al., 2003. Sex. Plant Repr. 16:141-9). The heat can produce a variation of this time, most likely a delay. The analyses conducted at 24h-48h at the latest (e.g. lines 115, 158, 238, 250) seems insufficient to make a proper assessment of the consequences of heat treatment. Variable results may be obtained depending on the stage in which the meiocytes are when heat treatment is applied.

We deliberately used relatively short periods of heat stress in our research study as this allows us to untangle distinct effects of heat on different relevant processes during meiotic cell division. Meiosis is highly complex requiring the integrated or subsequent action of many processes to guarantee the formation of four recombined haploid spores out of one diploid precursor. It is certainly possible that heat interferes with the ‘normal’ duration of the male meiotic cell cycle in Arabidopsis male sporogenesis, and thus that the heat treatments do not fully cover the whole meiotic cell cycle. It is equally possible that surrounding tissue including the tapetum influence the meiotic cycle progression and that these sporophytic cells are differentially sensitive to heat depending on their developmental stage. Our experiments were designed to make a detailed assessment of the consequences of heat stress on the genetics of spores. A variation of short heat treatment periods and subsequently cytology allowed the analysis of the impact on the meiotic outcome and development of resulting spores. We did that for many subsequent days to fully characterize the various effects of heat stress on different developmental stages and processes in Arabidopsis male meiosis.

Line 260 (results): assertions in relation to synapsis are not accompanied by quantification. In addition, to conclude the existence of a deficiency in the formation of the synaptonemal complex, DAPI staining is insufficient. It is necessary to perform an immunolocalisation to detect both ASY1 and ZYP1 proteins.

We do not completely agree with this remark. DAPI-stained chromosome spreads are used to assess for alterations in homolog pairing and synapsis as it distinctly allows for the discrimination between single (i.e. unsynapsed) and double (i.e. synapsed) chromosome threads. This is clearly demonstrated in the pachytene-stage meiocytes shown in Figure 5 (see arrows indicating unsynapsed genomic regions) and in Figure 6 (see detailed figure insets in Fig 6g, 6m and 6s). However, in order to meet with the request for immunostaining of relevant SC proteins, we have included a series of images that show the dynamics of ASY1 and ZYP1 under both normal conditions as well as heat stress. These immunocytology assays clearly show that chromosome-specific loading of ASY1 is not impaired or altered under heat. ZYP1 immunostaining of heat-stressed male meiocytes, on the other hand, often revealed genomic regions at

pachytene that are devoid of the SC transverse filament, indicating for local absence of SC formation. Hence, albeit not supported by quantitative analyses, these cytological observations support the notion that heat interferes with the formation of the synaptonemal complex and leads to genomic regions of asynapsis (and thus absence of cross-over formation).

Secondly, our observations suggest that heat-induced alterations in homolog synapsis occur ad hoc and thus randomly across the genome, making quantitative and correlative analyses of the spatial distribution of heat-induced asynapsis and shifts in CO establishment possibly non-relevant. Even if we would consider to do such analysis, it would require setting-up a technically challenging technology (probably involving a live imaging assays of dividing meiocytes with high-resolution microscopy to directly link SC alterations with changes in spatial CO designation). As this requires great technical expertise and is aimed at answering more complex questions, these analyses are out of the scope of presented research paper.

Line 267 (results): What do you mean with “individual chromosome units”? the frequency is not indicated (“a minor subset”). Apparently in the picture (Fig. 5m) the “unit” seems to be a fragment. It must be taken into account that heat treatment can affect chromatin and it can be more easily fragmented after the spreading protocol. These “fragments” may not be a direct consequence of the heat effect. Regarding the sentence “in later stages, no indications of chromosome fragmentation were retrieved” a clear fragment appears in Fig. 6v (telophase I).

Whether the chromosome thread (we called it chromosome unit to avoid conclusive nomination) is a fragment or an entire chromosome can not be determined based on the DAPI image. In this particular image fig5m, the DAPI stained thread is clearly separated from the pachytene cluster. This was not seen at regular temperature and occurred too infrequently to allow further assessment whether it is a fragment or a chromosome. If these structures resulted from unrepaired DSBs, we should have seen fragments at later stages. The dot in fig 6v can also be lagging or unattached chromosome. The low frequency (4%) at which these "unit" occur prevents a proper identification given the resources we have for our research.

The putative “promiscuous interactions” (line 274) between different chromosomes have to be confirmed by FISH, to ensure that they take place between non-homologous chromosomes.

In order to validate the promiscuous interactions between meiotic chromosomes upon exposure to heat stress we have included the analysis of haploid meiocytes (i.e. male meiosis in haploid Arabidopsis plants) and demonstrate the occurrence of non-homologous recombination. Since haploid plants contain only one set of chromosomes ($x = 1n = 5$), all five chromosomes in meiosis I are intrinsically different from each other and thus defined as non-homologous. Any physical interaction between two, three, or more chromosomes in a haploid PMC thus indicates for an interaction between non-homologous chromosomes and thus provides evidence for promiscuous CO interactions. Because of this, we do not think it is crucial to do additional FISH experiments. Performing FISH experiments at metaphase I are technically highly challenging.

The analysis of the CO frequency at different genomic intervals were made at 6, 7, 8 and 10 dpt. This means that the meiocytes analyzed started meiosis after heat treatment. This situation should be considered in the interpretation of the results.

*This remark is not completely correct. The FTL system used here to determine the impact of heat on meiotic CO frequency in Arabidopsis PMCs is based on the read-out of the segregation of fluorescent recombinant reporters (i.e. YFP, eCFP, etc.) in mature *qrt1-2-/-* tetrad-configured pollen grains that each originate from one single meiotic event. However, it is important to note that it approximately takes 5-9 days for a freshly generated spore by meiosis to develop into a mature pollen grain, and thus that the FTL segregation in mature pollen grains reflects the recombination pattern and frequency of meiocytes that occurred 6-10 days earlier. Hence, because of the lagging period between the actual impact on meiotic CO formation (i.e. prophase I) and the FTL-based read-out of the segregation fluorescent markers in mature pollen tetrad, we decided to monitor FTL segregation in mature pollen that are released 6-10 days post heat treatment. Since these analyses consistently show a shift in FTL segregation, and thus cross-over frequency, at 9 days post heat treatment (see bar plots in Figure 7), these analyses not only show a distinct effect of heat on meiotic CO distribution but also reveal that it takes approx. 9 days for a heat-stressed meiocyte to develop into a mature pollen grain.*

Line 410 (discussion): “Our findings therefore support the view that heat-induced alterations in CO homeostasis are caused by a desynchronization of CO establishment and meiotic cell cycle progression.” Since no analysis of the duration of meiosis is carried out this cannot be assumed.

We think that a discussion requires some speculation to inspire other scientists. The speculation is based on the analogy with the findings reported in the paper by Higgins et al suggesting that heat affects the synchronisation of homolog-based DSB repair.

Line 425 (discussion): “it is conceivable that heat induced shifts in CO patterning are not directly inferred by regulatory elements acting in trans, but instead are caused by cis factors such as altered chromatin status or DNA methylation patterning”. In the discussion there is a whole paragraph devoted to a possible effect on epigenetic marks. Although it might be a possibility in this study no experiment has been conducted to check for possible epigenetic changes. In my opinion, this part of the discussion should be reduced.

The speculation that epigenetic marks are possibly important elements in heat sensitivity of CO designation is debated at meetings and we think it is relevant in the context of this paper.

Minor comments:

- Please include the definition of SDR (Second Division Restitution) in the abstract.

A definition of SDR was included in the abstract

- The word “ectopic” appears several times throughout the text. Sometimes it refers to the recombination between non-homologous sequences, but other times it does not have a clear meaning (e.g. lines 46, 207, 235).

As the word ectopic does not provide a better understanding of the observations we have made, it has been deleted throughout the text.

- Please standardize the nomenclature for the mutants (e.g. *atspo11-1-3*^{-/-}, *sun1sun2*).

Because some mutants like atspo11-1-3^{-/-} are sterile, we prefer to indicate that the analysis was done using a homozygous mutant, indicated by the standard nomenclature : -/-

Reviewer #2 (Remarks to the Author):

The manuscript written by Nico and Geelen comprehensively demonstrates how temperature affects male meiotic restitution in Arabidopsis by convincing cytological and genetic approaches. First, they found that heat stress causes restitution of Arabidopsis male meiosis mainly due to the defects of cell wall formation in Meiosis II. Consistent observations of di- and polyploids from microspore early stage to mature pollen evidenced that heat-induced male meiotic restitution leads to higher ploidy gametes. In order to identify the cause, they experimented on centromere-linked FTL lines and Col-0 haploid plants to reveal that restitution primarily generates from SDR. In addition, they reported the cytological phenotype of heat-stressed meiocytes, which corresponded with CO alteration eventually, and inferred the occurrence of non-homologous recombination induced by heat.

This is a very nice piece of work with concise and well-organized results, and clear, straight-forward figures. However, authors should pay more attention to the details. There are more typos and mistakes in this manuscript than usual.

We have done our best to eliminate typos and mistakes.

Reviewers' comments:

Reviewer #1 (Remarks to the Author):

This study shows that heat stress affects the ploidy level of pollen and meiotic recombination in Arabidopsis using a combination of cytology and well-established genetic analyses. Two heat treatments (26-28 °C, 30-32 °C) of different durations were conducted, with a combination of different recovering time to proceed growth (i.e. various intervals after treatments until microscopic observation). First, the authors report that the different severity of heat stress induced different responses (Fig 1A). After various periods of time post treatment, authors showed tetrad stage was normal after 12 hours post treatment; instead, abnormality was observed at the microscope (Fig 1B). Aberrant cells can keep developing into pollen grains at 7-9 days post treatment as authors followed their development (Fig 3). Secondly, authors showed defects of RMAs and cell wall formation at the end of meiosis II (Fig 2). By analyzing fluorescence-tagged lines, authors further showed that heat stress primarily affects the SDR-type of meiotic restitution (76%), and the rest cases are FDR-type (Fig 4). Authors also applied heat stress to haploid plants and *atps1-1* mutant and strengthened the notion of increased SDR and FDR restitution under heat stress. Then, 6 hours and 24 hours- heat treatment (30-32C) resulted in incomplete synapsis and promiscuous interaction between nonhomologous chromosomes at diakinesis (Fig 5 and Fig 6). Lastly, authors showed altered CO distribution after 24h 30-32C treatment (Fig 7). In addition, the results from *spo11-1* mutant and haploid confirmed that ectopic chromosome interactions between non-homologous chromosomes depend on DSB formation (Fig 8).

This study presents a broad view of the impact of higher temperatures on meiotic events. Since the effects are pleiotropic, the authors have focused on how dyads are formed and how cross-overs are altered. The study is in general well documented. The findings regarding more polyads forming under higher temperature and more SDR-type meiotic restitution upon heat stress, as well as the transient effect of heat stress on the cross-over landscape, are informative and novel. However, authors report here a large amount of results with complicated experimental designs. In general, most experiments make sense to me, but the writing somehow is confusing and in a poor organization, thus difficult to follow. I believe there is still a large room for improvement. I would also suggest that the authors reorganize the order of their results because, in their current order, they are difficult to understand.

Major comments:

1. I suggest authors remove or trim some results of mild heat stress (26-28C), because most of results presented here were done with 30-32C treatment. Without detailed analysis of meiosis after 26-28C treatment, it is difficult to distinguish the exact impact by 26-28C or 30-32C. For example, incomplete synapsis, defects in recombination and RMAs can be also induced by 26-28C in lower frequency. Authors seem to state that defects of CO and synapsis are exclusively observed after 30-32C treatment. (Line 19)
2. The 26-28 °C treatment leads to the formation of dyads and triads, but the assays to determine SDR-type meiotic restitution and crossing-over frequency were conducted only after a higher temperature (30-32 °C) treatment. Note that the 30-32 °C treatment caused more severe and complicated defects since multiple mechanisms and regulatory pathways are potentially affected. Why not using 26-28 C treatment to determine SDR-type or FDR-type?
3. In the Introduction, the authors begin by stating that genome evolution is partially driven by polyploidy. However, in this study, no evidence is provided to demonstrate that these diploid pollen are viable or can transmit to generate 3N or 4N progeny. I suggest that the first part of the Introduction be rewritten.
4. Fig 1A shows that triad cells are the most common abnormality under any treatment regime. The authors should elaborate on this observation.
5. Please first introduce the time-course of Arabidopsis male gametogenesis. Otherwise, it is hard to follow which meiotic stage was subjected to heat treatment and difficult to understand how the authors come to the conclusion that "heat does not interfere with pre-meiotic PMC development" (Line 154-155).

6. I also suggest authors preparing a diagram showing the time course of treatments and recovering time as well as stages examined. That will help readers to understand better.

7. It will be informative to examine meiotic spindle organization at metaphase I and II after heat treatment. It is important because in my view, RMA MT biogenesis may not be altered. Instead, it may just follow the results of altered chromosome segregation due to abnormal spindle.

8. Line 358, it seems to me that there is no evidence or even suggestion of defective telomere clustering. Telomere positions would need to be carefully examined to make that claim.

Minor point:

1. Line 15, "26-29 °C" is not consistent with other parts of the paper.

2. Please reword the sentence "These ectopic chromosomal associations rely on DSB" (Line 23).

3. In the Introduction, the authors begin by stating that genome evolution is partially driven by polyploidy. However, in this study, no evidence is provided to demonstrate that these diploid pollen are viable or can transmit to generate 3N or 4N progeny. I suggest that the first part of the Introduction be rewritten.

4. remove the abbrev. "WGD" (line 42)

5. Line 49: extra "(".

6. Fig 1B, please indicate what "hpt" is in the figure legend.

7. Line 128-130, mild-heat treatment is mentioned in the text, but refers to Fig. 1B in which the results of 24 h treatment at 30-32 °C are presented.

8. Line 151: briefly describe the qrt mutant.

9. Fig 1B-t looks normal to me.

Reviewer #2 (Remarks to the Author):

In general, I am satisfied with the modifications made to the text and the replies received. I'm just going to point out a few details.

It would be necessary to include a reference to justify that the mature pollen is formed 6-7 days after meiosis.

Since the authors do not rule out that the chromosome unit in Figure 5m is a fragment, they cannot infer that there are failures in telomere clustering. I suggest the authors be more cautious with these descriptions.

Promiscuous interactions are validated by the analysis of haploid meiocytes. However, the bivalents that the authors mention are difficult to observe in the images (Figure 8B). Since this is a very interesting result and FISH has not been applied in the diploid material for analyzing interactions between non-homologous chromosomes, could the authors include an enlarged detail of any bivalent in the haploid?

Dear Yuan Qin, Editor Communications Biology,

In response to the comments on our revised manuscript entitled "Male meiotic restitution and altered cross-over distribution in *Arabidopsis thaliana* under heat" we have prepared a new version and prepared a rebuttal addressing the reviewers comments point by point.

An important issue raised by reviewer 1 was the request for additional experiments, which was also a concern that you emphasized in your letter. The question concerns the dependability of the meiosis defects on the level of heat stress, modest or very high temperature, and whether the alterations in meiotic recombination, RMA disorganization and frequency of SDR/FDR observed at 30-32°C also occur at the mild temperature stress of 26°C. Indeed, we may have given the impression that at least some of these defects are more specific for the elevated temperatures (30-32°C), without providing support from experimental data. The reason why we have not pursued a detailed analysis of meiosis at 26-28°C is simply because of the low frequency at which defects occur. The incidence of aberrant spore formation (at the end stage of meiosis) amounts, on average, to 5%, and encompasses formation of triads, dyads, unbalanced dyads and a small number of polyads. This fairly low, but significant number of defective meiocytes is the result of different chromosome segregation and cytokinesis defects.

The fact now is that we, and others, have reported the occurrence of a low number of dyads and triads (0-0,5%; there is variation between experiments) in wild type *Arabidopsis Col-0* plants grown at regular temperature conditions (De Storme and Geelen, *Plant Physiology* 2011; Erilova et al., *PLoS Genetics* 2009). Thus, to be statistically sound, detailed analyses of the meiotic phenotype at mild heat stress need to be determined with respect to a control that has not been exposed to a shift in temperature. Because of the low frequency of altered meiotic products at 26°C, detection of univalents and defects in RMA structure and cytokinesis have not been attempted. To distinguish "naturally" occurring errors in meiosis resulting in dyad formation from those induced by mild heat stress would require the recording of a very large number of events, for which we doubt will be feasible and is in our opinion beyond the scope of this paper.

We have however extensively studied the defects occurring at 32°C, and provide substantial illustration and data on the various types of changes heat stress causes in male meiocytes.

A noticeable difference between the mild and the more severe heat treatment is a disproportionately increase in polyad formation at the higher temperature. As polyads result from aberrant chromosome segregation, this is most likely caused by the absence of recombination in any given chromosome, resulting in MI univalent, chromosome laggards and aneuploidy. Again, laggard chromosomes and univalents also rarely occur in male meiocytes under regular temperature conditions rendering comparison with mildly stressed plants too demanding.

Two recent publications report on the impact of temperature on recombination frequency in *Arabidopsis* (Modliszewski et al., 2018, doi: 10.1371/journal.pgen.1007384; Lloyd et al., 2018, doi: 10.1534/genetics.117.300588). Here, in our study, we report that heat stress

increases the number of univalents at MI which results in an imbalance in chromosome segregation. As the ratio of polyad over triad and dyad formation increases at the high temperature condition (32°C), we think that reduced recombination and mis-segregation of chromosomes is the cause for polyad formation.

Response to reviewer 1:

Major comments:

1. I suggest authors remove or trim some results of mild heat stress (26-28C), because most of results presented here were done with 30-32C treatment. Without detailed analysis of meiosis after 26-28C treatment, it is difficult to distinguish the exact impact by 26-28C or 30-32C. For example, incomplete synapsis, defects in recombination and RMAs can be also induced by 26-28C in lower frequency. Authors seem to state that defects of CO and synapsis are exclusively observed after 30-32C treatment. (Line 19)

We have adapted the text to tune down the interpretation of mild heat stress and rewrote the discussion to remove over-interpretation of the mild heat stress result. The data on the impact of mild stress at 26C were moved to the supplemental section. The observation that Arabidopsis grown at 26C shows defects in male meiosis is very relevant for the Arabidopsis community and we therefore want to keep these data linked with our manuscript. We agree that most of our analysis was done exposing plants to higher temperature and that our conclusions should primarily be based on these experiments.

2. The 26-28 °C treatment leads to the formation of dyads and triads, but the assays to determine SDR-type meiotic restitution and crossing-over frequency were conducted only after a higher temperature (30-32 °C) treatment. Note that the 30-32 °C treatment caused more severe and complicated defects since multiple mechanisms and regulatory pathways are potentially affected. Why not using 26-28 C treatment to determine SDR-type or FDR-type?

The frequency at which balanced dyads occur (a prerequisite to monitor FDR/SDR type of restitution) under 26-28°C is in the order of 1-2 %. Under normal temperature conditions, wild type plants produce diploid pollen at a range between 0,01 and 0,5%, possibly depending on the levels of stress (water, drought, etc) they experience. This means that the difference between normal conditions and high temperature treatment is not very large and we would need to analyse many sample to obtain statistically sound data. In addition, the analysis is done using microscopic observations (not flow cytometry) requiring a significant effort to collect and analyse a large number of images (per field of view we may see up to 20 qrt spores).

3. In the Introduction, the authors begin by stating that genome evolution is partially driven by polyploidy. However, in this study, no evidence is provided to demonstrate that these diploid pollen are viable or can transmit to generate 3N or 4N progeny. I suggest that the first part of the Introduction be rewritten.

We endorse this criticism and rewrote the first part of the introduction. We want to point out to the reviewer that mutant lines producing 2n pollen, like jason, atps1, osd1, tam, etc generate significant numbers of triploid progeny. This is however not on a one to one basis because Arabidopsis Col0 shows triploid block, reducing the chances that a triploid embryo with tetraploid endosperm develops into a viable seed.

4. Fig 1A shows that triad cells are the most common abnormality under any treatment regime. The authors should elaborate on this observation.

Triad formation is indeed the most common defect at mild and severe temperature conditions. During cytokinesis, the four male haploid nuclei are kept at a distance from one another by radial microtubules, and generate a tetrahedral structure, which is the most compact organization possible. We observed the impact of heat stress by fixing cells and imaging the microtubules. These pictures are show in the paper. Clearly, some do not have microtubules separating the nuclei while other display microtubule structures that are not well organized as is typical in untreated tetrad spores. Since we cannot perform live imaging, we do not know the fate of the poorly structured RMA and whether it for instance disintegrates, to allow the nuclei to move close to one another at a later stage. The higher incidence to form triads, is in agreement with the observation that the heat induced defect does not originate from malfunctioning of the spindles in meiosis II as this would generate mainly dyads as in AtPS1 and Jason mutants.

5. Please first introduce the time-course of Arabidopsis male gametogenesis. Otherwise, it is hard to follow which meiotic stage was subjected to heat treatment and difficult to understand how the authors come to the conclusion that "heat does not interfere with pre-meiotic PMC development" (Line 154-155).

A similar remark was made by reviewer 2. The heat stress experiments were done with intact plants, moving them to a climate chamber set at an elevated temperature. At the moment of heat treatment, some of the flowers were pre-meiotic, meaning they were at a developmental stage before male meiosis. We found that these (smaller) flowers generated normal pollen and hence were not affected by heat.

6. I also suggest authors preparing a diagram showing the time course of treatments and recovering time as well as stages examined. That will help readers to understand better.

The flower buds were prepared for microscopic observation right after the heat treatments, except for data shown in figure 1, where it is explicitly indicated. The treatment time is shown in the image panels and the sampling time is spelled out in the materials and methods.

7. It will be informative to examine meiotic spindle organization at metaphase I and II after heat treatment. It is important because in my view, RMA MT biogenesis

may not be altered. Instead, it may just follow the results of altered chromosome segregation due to abnormal spindle.

In a previous study we have investigated the impact of cold stress on male microsporogenesis. In that study we found no chromosome dynamic defects and spindles appeared normal. Small changes (e.g. size) may have been overlooked but we did not see a change in the arrangement of microtubules like we observed for the RMA (De Storme et al., 2012). Upon heat stress, we did not systematically image the spindles. One should expect disorganized spindles in view of the meiotic spreads, but these are not per se linked with the formation of dyads. These altered spindles would confound the changes that would be responsible for dyad or triad formation and complicate such analysis profoundly. An argument why we think that spindle defects are not the cause for dyad formation is that these changes lead to FDR type of restitution. The parallel and tripolar spindles produced by for instance the male meiosis mutants *Jason* and *ps1*, generate FDR-type meiotic restitution, whereas heat causes primarily SDR type of restitution.

Our interpretation of the results is that temperature stress causes the microtubule network to break down and when the temperature returns to normal, the microtubules repolymerize and reform a network. In the tetrads this involves polymerization in between the nuclei that keeps them separated. In those cases that nuclei in close proximity, nucleation of microtubules could not take place and the nuclei are perceived as a single entity. Hence, separation is no longer possible. Since a tetrad stage meiocyte will more likely carry 2 nuclei that are close to one another than that 2 times 2 nuclei have moved towards each other, we expect to find a higher incidence of triads compared to dyads. This is different in the spindle organization mutants *AtPS1* and *Jason*, where we find a higher number of dyads over triads. In the double *PS1/Jas* mutant 85% of the spores are dyads (De storme et al., 2011).

8. Line 358, it seems to me that there is no evidence or even suggestion of defective telomere clustering. Telomere positions would need to be carefully examined to make that claim.

We agree we do not provide any evidence for our suggestion and have removed the possibility of involvement of telomere clustering.

Minor point: These have all been addressed.

1. Line 15, "26-29 °C" is not consistent with other parts of the paper.
2. Please reword the sentence "These ectopic chromosomal associations rely on DSB" (Line 23).
3. In the Introduction, the authors begin by stating that genome evolution is partially driven by polyploidy. However, in this study, no evidence is provided to demonstrate that these diploid pollen are viable or can transmit to generate 3N or 4N progeny. I suggest that the first part of the Introduction be rewritten.
4. remove the abbrev. "WGD" (line 42)
5. Line 49: extra "(".
6. Fig 1B, please indicate what "hpt" is in the figure legend.

7. Line 128-130, mild-heat treatment is mentioned in the text, but refers to Fig. 1B in which the results of 24 h treatment at 30-32 °C are presented.
8. Line 151: briefly describe the qrt mutant.
9. Fig 1B-t looks normal to me.

Reviewer #2 (Remarks to the Author):

1. It would be necessary to include a reference to justify that the mature pollen is formed 6-7 days after meiosis.

We have added a figure to indicate the time line of meiosis and spore development. In addition we refer to our previous work where cold treatment follows a similar timeline, affecting the RMA during meiosis and resulting in mature 2n spores 7 days later.

2. Since the authors do not rule out that the chromosome unit in Figure 5m is a fragment, they cannot infer that there are failures in telomere clustering. I suggest the authors be more cautious with these descriptions.

Agreed, cfr reviewer 1.

3. Promiscuous interactions are validated by the analysis of haploid meiocytes. However, the bivalents that the authors mention are difficult to observe in the images (Figure 8B). Since this is a very interesting result and FISH has not been applied in the diploid material for analyzing interactions between non-homologous chromosomes, could the authors include an enlarged detail of any bivalent in the haploid?

An inset image is included showing a blow up of the relevant part of the meiotic spread. Indeed, the observation has interesting implications for the mechanisms recombination in the absence of homology. However, regarding the FISH experiment, in our opinion this will not give more insights as it is based on the same meiotic spreading technique, and it will show that the haploid inducer indeed generates spores lacking chromosomes. Further research would require the analysis of mutant lines under heat stress conditions, which is beyond the scope of this paper.